# Superconducting Phases in Neutron Star Cores

Toby S. Wood [1,*] and Vanessa Graber [2,3]

1    School of Mathematics, Statistics and Physics, Newcastle University, Newcastle upon Tyne NE1 7RU, UK
2    Institute of Space Sciences (ICE-CSIC), Campus UAB, 08193 Barcelona, Spain; graber@ice.csic.es
3    Institut d'Estudis Espacials de Catalunya (IEEC), 08034 Barcelona, Spain
*    Correspondence: toby.wood@newcastle.ac.uk

**Abstract:** Using a phenomenological Ginzburg–Landau model that includes entrainment, we identify the possible ground states for the neutron and proton condensates in the core of a neutron star, as a function of magnetic field strength. Combining analytical and numerical techniques, we find that much of the outer core is likely to be a "type-1.5" superconductor (instead of a type-II superconductor as often assumed), in which magnetic flux is distributed inhomogeneously, with bundles of magnetic fluxtubes separated by flux-free Meissner regions. We provide an approximate criterion to determine the transition between this type-1.5 phase and the type-I region in the inner core. We also show that bundles of fluxtubes can coexist with non-superconducting regions, but only in a small part of the parameter space.

**Keywords:** neutron stars; superconductivity; superfluidity

## 1. Introduction

In the core of a neutron star, neutrons and protons are both expected to form condensates via Cooper pairing, leading to a neutron superfluid and a proton superconductor, respectively. Within the proton superconductor magnetic flux is quantized into microscopic filaments called fluxtubes, and the macroscopic dynamics of the star's core magnetic field depends on the microscopic dynamics of these fluxtubes. In the conventional picture [1–3], the inner core is a type-I superconductor, meaning that fluxtubes are mutually attractive and therefore coalesce into macroscopic patches of magnetic flux, whereas the outer core is a type-II superconductor, meaning that fluxtubes are mutually repulsive and therefore form a regular array. The transition between these two superconducting regimes occurs where the ratio of the London penetration length, $\lambda$, and proton coherence length, $\xi_p$, the so-called Ginzburg–Landau parameter $\kappa \equiv \lambda/\xi_p$, takes the value $\kappa = 1/\sqrt{2}$. The inner and outer core are therefore defined to be the regions in which $\kappa < 1/\sqrt{2}$ and $\kappa > 1/\sqrt{2}$, respectively. However, the microscopic behavior of fluxtubes in the core is affected by the coupling between the proton and neutron condensates, and in particular by their mutual entrainment, which could significantly change the location of this transition, and might even result in entirely different types of superconductivity. For example, Buckley et al. [4,5] showed that strong density coupling between the condensates could produce type-I superconductivity throughout the whole core, although only if the coupling is much stronger than is generally expected [6]. A more detailed study by Alford and Good [7], incorporating density and density-gradient coupling, found that in some cases the transition from type-I to type-II is mediated by domains of "type-II($n$)" superconductivity, wherein each fluxtube carries $n$ magnetic flux quanta. Subsequently, Haber and Schmitt [8] argued that these type-II($n$) fluxtubes are generally unstable, and instead there is a regime of "type-1.5" superconductivity, in which the fluxtubes form bundles with a preferred separation. Such unconventional behavior has also been discussed in the context of terrestrial multi-band superconductors [9–13]. We will show later that inhomogeneous flux distributions arise in our model under a wide range of parameter conditions, as a result of coupling between the two

condensates. We adopt the term "type-1.5" superconductivity to maintain consistency with earlier literature on the topic. We note that there are other mechanisms that might produce bundles of fluxtubes inside neutron stars, such as a macroscopic instability of the fluxtube lattice [14] or interactions with quantized vortices in the neutron superfluid [15]. However, these scenarios do not correspond to type-1.5 superconductivity in the sense used here.

The nature of superconductivity in the star's core may have observational consequences. For example, there is observational evidence for long-period precession in some neutron stars [16–19], which places constraints on dissipation within the core. If the superfluid neutron vortices in the outer core are tightly pinned to a regular array of fluxtubes then precession becomes essentially impossible [20,21]. This contradiction could be resolved if the entire core were in a type-I state [21–23], or if instabilities resulting from precession are able to unpin the vortices [24]. Pinning between vortices and fluxtubes could also significantly affect the rise time, amplitude, and relaxation rate of rotational glitches in pulsars [25–27], as well as the long-term evolution of their rotation and magnetic field [28,29].

The goal of this paper is to extend previous studies to include all relevant couplings between the neutron and proton condensates, and to determine the precise conditions under which different types of superconductivity arise. We use a Ginzburg–Landau model for the condensates, which is the simplest phenomenological description that captures the quantization of vorticity and magnetic flux. Specifically, we work with the most general Ginzburg–Landau functional that permits a consistent treatment of entrainment, while also correctly satisfying Galilean invariance on small scales [30,31]. Using analytical and numerical techniques, we then find the ground state for the condensates by minimizing the free energy of the two-component system in the presence of an imposed magnetic field, and thus construct superconducting phase diagrams. Although our aim here is only to identify the ground state, and hence to determine the type of superconductivity, we note that this *local* ground state is likely to be a good approximation to the situation in real neutron stars on small scales. This is because young neutron stars cool very efficiently via neutrino emission [32,33], and after $\sim 10^4$ yr their internal temperatures lie far below the critical temperatures for superconductivity and superfluidity [34–36].

The paper is organized as follows. In Section 2 we present our Ginzburg–Landau model, including a review of entrainment and Galilean invariance. In Section 3 we solve the Ginzburg–Landau equations numerically, and hence construct superconducting phase diagrams. Section 4 presents analytical results concerning the various phase transitions found numerically. Finally, we discuss the implications for neutron stars in Section 5. Further details of some of the calculations are provided in Appendices A–C.

## 2. The Ginzburg–Landau Formalism

Our goal is to formulate a simple, phenomenological model of the neutron and proton condensates that includes (a) their mutual entrainment, and (b) the coupling to the magnetic field. The simplest such model is the Ginzburg–Landau model, in which the free energy of the condensates is expressed in terms of complex scalar order parameters for the proton and neutron condensates, $\psi_{\rm p}$ and $\psi_{\rm n}$, and the magnetic vector potential, **A**. We acknowledge that such a model can only be rigorously justified close to the condensation temperatures for the protons and neutrons, whereas mature neutron star cores are generally well below both of those temperatures. This limits the rigorous applicability of the Ginzburg–Landau model to the part of the core in which the condensation temperatures are similar [37], and to the narrow time window in which the condensation occurs. Furthermore, the Ginzburg–Landau model neglects spin–orbit interactions, and therefore does not include all of the microphysics of a neutron star. In particular, the neutrons are believed to form an anisotropic $^3P_2$ superfluid throughout much of the core, which cannot be described by a single scalar order parameter, e.g., [38]. Nevertheless, the relative simplicity of the Ginzburg–Landau formalism makes it an appealing phenomenological model, and hence it has been widely used to model the superfluid components of neutron stars, e.g., [7,8,39–43]. Moreover, phenomenological Ginzburg–Landau models have been shown to reproduce many properties of laboratory

superconductors, even well below the condensation temperature [44]. We will therefore take a similar approach to model the neutron star core.

In what follows, we normalize the order parameters such that $|\psi_\mathrm{p}|^2$, for example, is the number density of proton Cooper pairs, and so $\rho_\mathrm{p} = 2m_\mathrm{p}|\psi_\mathrm{p}|^2$ is the mass density of the proton condensate, where $m_\mathrm{p}$ is the mass of a proton. The order parameters are therefore two-particle mean-field wave functions for the condensates. As noted earlier, the core temperature in mature neutron stars lies far below the critical temperature for superconductivity, so in the absence of magnetic flux practically all of the proton matter would reside in the condensed state. In the presence of magnetic flux, however, there will be normal proton matter present in the cores of fluxtubes and in any non-superconducting regions, where the proton condensate is absent. We are concerned here only with the ground state for the condensates, wherein interactions with any "normal" components of the core, which include electrons, thermal excitations and normal protons and neutrons, are suppressed. Hence, we can safely disregard the normal matter in what follows.

*2.1. Entrainment and Local Phase Invariance*

The residual strong interaction between neutrons and protons in the core leads to a non-dissipative drag between their condensates known as entrainment [39,45]. To illustrate the dynamical effect of entrainment, we temporarily neglect any coupling to the magnetic field by setting $\mathbf{A} = \mathbf{0}$; the magnetic field will be reintroduced later by invoking gauge invariance. The hydrodynamical momenta of the condensates are then proportional to the gradient of the phases of the order parameters, i.e.,

$$\hbar\boldsymbol{\nabla}\arg\psi_\mathrm{p} = 2m_\mathrm{p}\mathbf{V}_\mathrm{p}\,, \qquad \hbar\boldsymbol{\nabla}\arg\psi_\mathrm{n} = 2m_\mathrm{n}\mathbf{V}_\mathrm{n}\,, \tag{1}$$

where $\mathbf{V}_\mathrm{p}$ and $\mathbf{V}_\mathrm{n}$ are the superfluid velocities. In the presence of entrainment, the velocity-dependent terms in the free-energy density, $F_\mathrm{vel}$, of the condensates must take the form

$$F_\mathrm{vel} = \tfrac{1}{2}\rho_\mathrm{p}|\mathbf{V}_\mathrm{p}|^2 + \tfrac{1}{2}\rho_\mathrm{n}|\mathbf{V}_\mathrm{n}|^2 - \tfrac{1}{2}\rho^\mathrm{pn}|\mathbf{V}_\mathrm{p} - \mathbf{V}_\mathrm{n}|^2\,, \tag{2}$$

where $\rho_\mathrm{p}$ and $\rho_\mathrm{n}$ are the true mass densities of the condensates and the coefficient $\rho^\mathrm{pn}$, which determines the strength of entrainment, is generally negative [46]. This form of the free energy—with an interaction term that depends only on the relative velocity—is necessary to ensure Galilean invariance [31] (as well as the more general constraint of "local phase invariance" [30]).

Equation (2) demonstrates that the primary effect of entrainment is to disfavor any relative flow between the two condensates by imposing an energetic penalty wherever $\mathbf{V}_\mathrm{p} \neq \mathbf{V}_\mathrm{n}$. A more subtle but equally important consequence is that the condensates' hydrodynamical momenta, given in Equation (1), are no longer proportional to their mass fluxes, defined as $\partial F_\mathrm{vel}/\partial\mathbf{V}_x$ for $x \in \{\mathrm{p},\mathrm{n}\}$. This has significant consequences for the structure of fluxtubes and vortices, and for their mutual interactions [39], but in the present work we are concerned only with fluxtubes, i.e., topological defects in the proton condensate. For this reason, we will neglect the star's rotation, meaning that no neutron vortices are present in the ground state. While neglecting rotation is certainly a simplification, it is justified when deriving a microscale model of the neutron star interior, in which the proton fluxtube density is many orders of magnitude larger than that of the neutron vortices, e.g., [47].

Within the Ginzburg–Landau mean-field framework, entrainment first enters the free-energy density at fourth order in the order parameters, and at second order in their derivatives [39]. The most general such term that satisfies global U(1) symmetry in each condensate is a linear combination of the quantities

$$|\psi_x|^2|\boldsymbol{\nabla}\psi_y|^2,\ \psi_x\psi_y\boldsymbol{\nabla}\psi_x^\star\cdot\boldsymbol{\nabla}\psi_y^\star,\ \psi_x\psi_y^\star\boldsymbol{\nabla}\psi_x^\star\cdot\boldsymbol{\nabla}\psi_y,\ \psi_x^\star\psi_y^\star\boldsymbol{\nabla}\psi_x\cdot\boldsymbol{\nabla}\psi_y, \tag{3}$$

where $x,y \in \{\mathrm{p},\mathrm{n}\}$ and $\star$ indicates the complex conjugate. With the additional constraint of Galilean invariance (2), the most general form of the entrainment term is found to be

$$F_{\text{ent}} = \tfrac{1}{2}(h_1 + h_2)\left|\left(\frac{m_{\text{n}}}{m_{\text{p}}}\right)^{1/2}\psi_{\text{n}}^{\star}\boldsymbol{\nabla}\psi_{\text{p}} + \left(\frac{m_{\text{p}}}{m_{\text{n}}}\right)^{1/2}\psi_{\text{p}}\boldsymbol{\nabla}\psi_{\text{n}}^{\star}\right|^2$$

$$+ \tfrac{1}{2}(h_1 - h_2)\left|\left(\frac{m_{\text{n}}}{m_{\text{p}}}\right)^{1/2}\psi_{\text{n}}\boldsymbol{\nabla}\psi_{\text{p}} - \left(\frac{m_{\text{p}}}{m_{\text{n}}}\right)^{1/2}\psi_{\text{p}}\boldsymbol{\nabla}\psi_{\text{n}}\right|^2$$

$$+ \tfrac{1}{4}h_3\left|\boldsymbol{\nabla}(\psi_{\text{p}}\psi_{\text{p}}^{\star})\right|^2 + \tfrac{1}{4}h_4\left|\boldsymbol{\nabla}(\psi_{\text{n}}\psi_{\text{n}}^{\star})\right|^2, \tag{4}$$

which includes four real, independent parameters $h_1, \ldots, h_4$. In terms of the superfluid densities and velocities, we can write this as

$$F_{\text{ent}} = h_1\left[\frac{\rho_{\text{n}}}{4m_{\text{p}}^2}\left|\boldsymbol{\nabla}\rho_{\text{p}}^{1/2}\right|^2 + \frac{\rho_{\text{p}}}{4m_{\text{n}}^2}\left|\boldsymbol{\nabla}\rho_{\text{n}}^{1/2}\right|^2 + \frac{\rho_{\text{p}}\rho_{\text{n}}}{\hbar^2}|\mathbf{V}_{\text{p}} - \mathbf{V}_{\text{n}}|^2\right]$$

$$+ \frac{h_2}{8m_{\text{p}}m_{\text{n}}}\boldsymbol{\nabla}\rho_{\text{p}} \cdot \boldsymbol{\nabla}\rho_{\text{n}} + \frac{h_3}{16m_{\text{p}}^2}|\boldsymbol{\nabla}\rho_{\text{p}}|^2 + \frac{h_4}{16m_{\text{n}}^2}|\boldsymbol{\nabla}\rho_{\text{n}}|^2. \tag{5}$$

So by comparison with Equation (2) the entrainment coefficient is

$$\rho^{\text{pn}} = -\frac{2}{\hbar^2}h_1\rho_{\text{p}}\rho_{\text{n}}. \tag{6}$$

The parameters $h_2, h_3, h_4$ only provide density-gradient coupling, and the simplest model of entrainment would therefore set these parameters to zero. On scales much larger than the fluxtube cores the superfluid densities are approximately constant, and so these terms will have negligible effect. However, we will show later that the density-gradient terms play a significant role in the transition between type-I and type-II superconductivity, and therefore must be included in the construction of phase diagrams.

*2.2. Connection with Previous Work*

Our general expression (5) for the entrainment energy differs from corresponding expressions found in some previous works. To highlight the differences, we note that the velocity contributions to the free-energy density, given by Equation (2), can equivalently be expressed as

$$F_{\text{vel}} = \tfrac{1}{2}\rho^{\text{pp}}|\mathbf{V}_{\text{p}}|^2 + \tfrac{1}{2}\rho^{\text{nn}}|\mathbf{V}_{\text{n}}|^2 + \rho^{\text{pn}}\mathbf{V}_{\text{p}} \cdot \mathbf{V}_{\text{n}}, \tag{7}$$

where $\rho^{\text{pp}} \equiv \rho_{\text{p}} - \rho^{\text{pn}}$ and $\rho^{\text{nn}} \equiv \rho_{\text{n}} - \rho^{\text{pn}}$ represent "effective" proton and neutron mass densities. Therefore, on scales much larger than the vortex and fluxtube cores, for which the superfluid densities are approximately constant, entrainment can be described by including in the free-energy density a term proportional to $\mathbf{V}_{\text{p}} \cdot \mathbf{V}_{\text{n}}$, and renormalizing the proton and neutron masses accordingly. However, in a microscale model that correctly includes density variations in the fluxtube cores, the dependence of the coefficients $\rho^{xy}$ on the condensate densities must be chosen carefully to preserve Galilean invariance [31,37] and additional density-gradient coupling terms have to be included. A number of previous studies, e.g., [39,43] do not treat entrainment on small scales consistently, because their entrainment interactions are incompatible with Equations (4) and (5). Haber and Schmitt [8] have introduced a relativistic model of density and derivative couplings that in the non-relativistic limit is also incompatible with our Equation (5), cf. their Equation (5). The model of Alford and Good [7] *is* compatible with Equations (4) and (5), but it only includes the $h_2$ term. Hence their model actually has no entrainment at all, i.e., $\rho^{\text{pn}} = 0$. This appears to be an oversight on their part, because they chose the value for $h_2$ based on prior estimates of $\rho^{\text{pn}}$. Finally, the model of Kobyakov [37] has a similar but subtly different form to Equation (4), because it takes the entrainment term to be

$$\frac{\left(\text{Im}\{|\psi_{\text{n}}|^2\psi_{\text{p}}^{\star}\boldsymbol{\nabla}\psi_{\text{p}} - |\psi_{\text{p}}|^2\psi_{\text{n}}^{\star}\boldsymbol{\nabla}\psi_{\text{n}}\}\right)^2}{|\psi_{\text{p}}|^2|\psi_{\text{n}}|^2}. \tag{8}$$

Although this quantity is Galilean invariant, it cannot be obtained from products of the order parameters, their conjugates and derivatives, and therefore cannot arise in our mean-field formalism. For the same reason, our model does not include a term proportional to $\boldsymbol{\nabla}|\psi_p| \cdot \boldsymbol{\nabla}|\psi_n|$, unlike the model of Kobyakov [37]. In what follows, we highlight where our results reproduce those of earlier studies, in appropriate limits.

## 3. Superconducting Ground States

### 3.1. The Free Energy Density

The total free-energy density in our model is obtained by adding the entrainment terms (4) to the usual free energy of a two-component superfluid, and introducing the magnetic vector potential **A** by minimal coupling. To reduce the number of parameters in the model, in what follows we assume that the protons and neutrons have equal masses, $m_p = m_n = m_u$, and that $h_3 = h_4$, which means that the entrainment interaction (4) is symmetric in the condensates. Using Gaussian c.g.s. units, the free energy density in its most compact form can then be expressed as

$$F[\psi_p, \psi_n, \mathbf{A}] = \frac{g_{pp}}{2}\left(|\psi_p|^2 - \frac{n_p}{2}\right)^2 + \frac{g_{nn}}{2}\left(|\psi_n|^2 - \frac{n_n}{2}\right)^2$$

$$+ g_{pn}\left(|\psi_p|^2 - \frac{n_p}{2}\right)\left(|\psi_n|^2 - \frac{n_n}{2}\right) + \frac{1}{8\pi}|\boldsymbol{\nabla} \times \mathbf{A}|^2$$

$$+ \frac{\hbar^2}{4m_u}\left|\left(\boldsymbol{\nabla} - \frac{2ie}{\hbar c}\mathbf{A}\right)\psi_p\right|^2 + \frac{\hbar^2}{4m_u}|\boldsymbol{\nabla}\psi_n|^2$$

$$+ h_1\left|\left(\boldsymbol{\nabla} - \frac{2ie}{\hbar c}\mathbf{A}\right)(\psi_n^\star\psi_p)\right|^2 + \frac{h_2 - h_1}{2}\boldsymbol{\nabla}(|\psi_p|^2) \cdot \boldsymbol{\nabla}(|\psi_n|^2)$$

$$+ \frac{h_3}{4}\left(\left|\boldsymbol{\nabla}(|\psi_p|^2)\right|^2 + \left|\boldsymbol{\nabla}(|\psi_n|^2)\right|^2\right), \tag{9}$$

where we have assumed that the proton Cooper pairs have charge $2e$. The coefficients $g_{pp}$ and $g_{nn}$ measure the self-repulsion of the condensates, and $g_{pn}$ measures their mutual repulsion or attraction. In the absence of magnetic field (i.e., if $\mathbf{B} \equiv \boldsymbol{\nabla} \times \mathbf{A} = \mathbf{0}$) we expect the ground state to be a uniform mixture of proton and neutron condensates, with position-independent densities $|\psi_p|^2 = n_p/2$ and $|\psi_n|^2 = n_n/2$. Therefore the parameters $n_p$ and $n_n$ represent the expected number density of superfluid protons and neutrons, respectively, in the absence of magnetic field. For reasons explained earlier, in a mature neutron star these will have values very close to the total (i.e., normal plus condensate) particle number densities. We will hereafter refer to this state, with uniform condensate densities and vanishing magnetic field, as the Meissner state, and we note that it has $F = 0$, according to Equation (9). However, if the mutual attraction/repulsion between the condensates is too strong, such that $g_{pn}^2 > g_{pp}g_{nn}$, then this Meissner state becomes unstable. Such behavior is not expected in the neutron star core, where the two condensates are believed to be only weakly attractive [6], and so in what follows we will always assume that $g_{pn}^2 < g_{pp}g_{nn}$. Moreover, we expect a neutron condensate to be present even in non-superconducting regions, where $\psi_p = 0$, and this implies the further restriction $g_{pn} > -g_{nn}n_n/n_p$. The consequences of violating these restrictions have been discussed in detail by Haber and Schmitt [8], for instance. In most studies of two-component condensates, the coefficient $g_{pn}$ represents the principle interaction between the two components, e.g., [48–51], and its effect on superconductivity in the neutron star core has been studied extensively [7,8,37]. In the present work, however, our main focus is on the effect of entrainment and other higher-order coupling terms. In the numerical results we present later, we therefore take $g_{pn} = 0$, and instead study the effect of the $h_i$ parameters on the superconductor. For completeness, and to facilitate comparison with earlier studies, we retain $g_{pn}$ in all of our analytical results.

In the absence of coupling between the condensates (i.e., for $g_{pn} = 0$ and $h_i = 0$) the "bare" coherence lengths are defined as

$$\xi_{\rm p} \equiv \frac{\hbar}{\sqrt{2m_{\rm u}g_{\rm pp}n_{\rm p}}} \quad \text{and} \quad \xi_{\rm n} \equiv \frac{\hbar}{\sqrt{2m_{\rm u}g_{\rm nn}n_{\rm n}}} \,, \tag{10}$$

and the "bare" London length is defined as

$$\lambda \equiv \sqrt{\frac{m_{\rm u}c^2}{4\pi{\rm e}^2 n_{\rm p}}} \,. \tag{11}$$

We will describe later how the effective coherence lengths and London length are modified by the coupling between the condensates, including their mutual entrainment.

In order to simplify the mathematical model, we now nondimensionalize the free-energy density (9) by measuring $\psi_{\rm p}$ and $\psi_{\rm n}$ in units of $\sqrt{n_{\rm p}/2}$ and $\sqrt{n_{\rm n}/2}$, respectively, lengths in units of $\xi_{\rm p}$, $\mathbf{A}$ in units of $\hbar c/(2{\rm e}\xi_{\rm p})$, and the coupling coefficients $h_i$ in units of $g_{\rm pp}\xi_{\rm p}^2$. To improve the readability of our equations, we avoid introducing specific notation for dimensionless quantities and instead point out that, from here on, all parameters refer to dimensionless quantities. The dimensionless free-energy density is then, in units of $g_{\rm pp}n_{\rm p}^2/4$,

$$
\begin{aligned}
F[\psi_{\rm p}, \psi_{\rm n}, \mathbf{A}] &= \left|(\boldsymbol{\nabla} - {\rm i}\mathbf{A})\psi_{\rm p}\right|^2 + \frac{1}{\epsilon}|\boldsymbol{\nabla}\psi_{\rm n}|^2 + \kappa^2|\boldsymbol{\nabla} \times \mathbf{A}|^2 \\
&+ \frac{1}{2}(1 - |\psi_{\rm p}|^2)^2 + \frac{R^2}{2\epsilon}(1 - |\psi_{\rm n}|^2)^2 + \frac{\alpha}{\epsilon}(1 - |\psi_{\rm p}|^2)(1 - |\psi_{\rm n}|^2) \\
&+ \frac{h_1}{\epsilon}\left|(\boldsymbol{\nabla} - {\rm i}\mathbf{A})(\psi_{\rm n}^\star\psi_{\rm p})\right|^2 + \frac{(h_2 - h_1)}{2\epsilon}\boldsymbol{\nabla}(|\psi_{\rm p}|^2) \cdot \boldsymbol{\nabla}(|\psi_{\rm n}|^2) \\
&+ \frac{h_3}{4}\left(\left|\boldsymbol{\nabla}(|\psi_{\rm p}|^2)\right|^2 + \frac{1}{\epsilon^2}\left|\boldsymbol{\nabla}(|\psi_{\rm n}|^2)\right|^2\right),
\end{aligned}
\tag{12}
$$

where we have defined the following parameters:

$$\kappa \equiv \frac{\lambda}{\xi_{\rm p}}, \quad R \equiv \frac{\xi_{\rm p}}{\xi_{\rm n}}, \quad \epsilon \equiv \frac{n_{\rm p}}{n_{\rm n}}, \quad \alpha \equiv \frac{g_{\rm pn}}{g_{\rm pp}}. \tag{13}$$

Note that $\kappa$ is equivalent to our dimensionless "bare" London length.

### 3.2. The Helmholtz and Gibbs Free Energies

We now seek the ground state for this system in the presence of an imposed magnetic field. There are two distinct thought-experiments that can be considered. In the first experiment, we control the magnetic flux density, $\mathbf{B}$, by imposing a mean or net magnetic flux, and minimize the Helmholtz free energy,

$$\mathcal{F} = \langle F \rangle, \tag{14}$$

where the angled brackets represent some kind of integral over our physical domain, which could be finite or infinite. This experiment closely approximates the conditions in the core of a neutron star, which becomes superconducting as the star cools in the presence of a pre-existing magnetic flux. However, as we will discuss below, the ground state under these conditions can be inhomogeneous, i.e., macroscopic domains of distinct physical behavior can appear. For conceptual convenience, we can consider an alternative experiment in which the system is coupled to a thermodynamic external magnetic field, $\mathbf{H}$, by minimizing the dimensionless Gibbs free energy,

$$\mathcal{G} = \langle F - 2\kappa^2\mathbf{H} \cdot \boldsymbol{\nabla} \times \mathbf{A} \rangle = \mathcal{F} - 2\kappa^2\mathbf{H} \cdot \langle \mathbf{B} \rangle. \tag{15}$$

In an unbounded domain, the ground state in this experiment is guaranteed to be homogeneous, and hence the phase diagram is generally simpler. For later reference, we present in Figure 1 the phase diagrams for a single-component Ginzburg–Landau

superconductor, i.e., we discuss its state as a function of the Ginzburg–Landau parameter, $\kappa$. (For more details, we refer the reader to standard textbooks on superconductivity, e.g., Tinkham [52]). For $\kappa < 1/\sqrt{2}$, we have a type-I superconductor; when **H** is used as the control parameter, there is a first-order transition between the Meissner state (with **B** = **0**) and the non-superconducting state (with **B** = **H**) at the critical value $|\mathbf{H}| = H_c = 1/(\sqrt{2}\kappa)$ in our dimensionless units. When the mean magnetic flux, $\overline{B}$, is used as the control parameter, this discontinuity resolves into an intermediate phase for $0 < \overline{B} < H_c$, in which Meissner regions alternate with non-superconducting ones. For $\kappa > 1/\sqrt{2}$, on the other hand, we have a type-II superconductor; for $H_{c1} < |\mathbf{H}| < H_{c2}$ the magnetic flux organizes into a hexagonal lattice of discrete fluxtubes. The transitions at the lower critical field, $H_{c1}$, and the upper critical field, $H_{c2}$, are both second-order, because fluxtubes appear with infinite separation at $|\mathbf{H}| = H_{c1}$, and the superconductor density becomes vanishingly small at $|\mathbf{H}| = H_{c2}$. In our dimensionless units, $H_{c2} = 1$ and $H_{c1} = \mathcal{F}_\infty/(4\pi\kappa^2)$, where $\mathcal{F}_\infty$ is the energy per unit length of a single fluxtube. When $\overline{B}$ is the control parameter, there is a similar second-order transition at $\overline{B} = H_{c2}$, and a first-order transition between the intermediate and fluxtube states at $\kappa = 1/\sqrt{2}$.

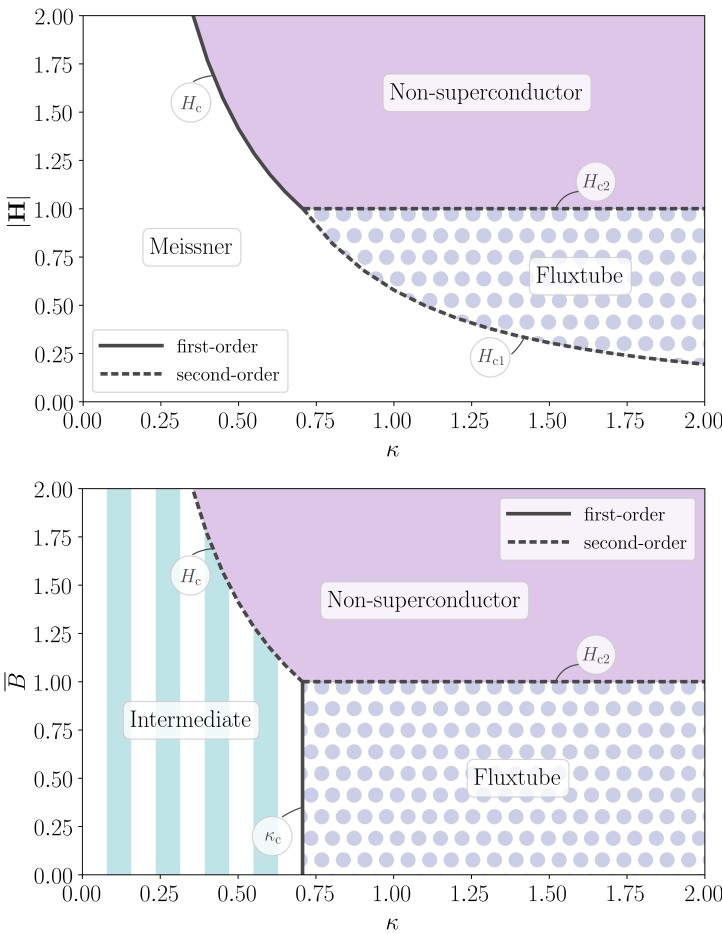

**Figure 1.** Phase diagrams for a one-component Ginzburg–Landau superconductor, for different values of the Ginzburg–Landau parameter, $\kappa$. The top panel shows the experiment with an imposed external field, $|\mathbf{H}|$, in our nondimensional units. The first-order and second-order transitions at the different critical fields are indicated by solid and dashed black lines, respectively, and the resulting phases labeled accordingly. Shading of the respective regions is indicative of the magnetic flux distribution. The bottom panel shows the phase transitions in the experiment with an imposed mean flux, $\overline{B}$. For more details see the text.

For our two-component system, we anticipate that the phase diagram will be more complicated than that shown in Figure 1. In particular, Haber and Schmitt [8] have argued that the upper and lower transitions to and from the fluxtube state can become first-order in some cases, occurring at $|\mathbf{H}| = H_{c1'} < H_{c1}$ and $|\mathbf{H}| = H_{c2'} > H_{c2}$, respectively. In that case, in the experiment with an imposed mean magnetic flux, $\overline{B}$, the ground state can feature an irregular array of fluxtubes, even in an unbounded domain. Some aspects of this phase space can be determined analytically, as we describe in Section 4. However, in order to produce a complete phase diagram, it is necessary to solve the Euler–Lagrange equations arising from one of the functionals $\mathcal{F}$ or $\mathcal{G}$ numerically, as we describe in the next section.

*3.3. The Numerical Model*

Whether we choose to work with the Helmholtz free energy, $\mathcal{F}$, or with the Gibbs free energy, $\mathcal{G}$, we obtain the same system of Euler–Lagrange equations:

$$\kappa^2 \boldsymbol{\nabla} \times (\boldsymbol{\nabla} \times \mathbf{A}) = \text{Im}\left\{ \psi_\text{p}^\star (\boldsymbol{\nabla} - \text{i}\mathbf{A})\psi_\text{p} + \frac{h_1}{\epsilon}\psi_\text{n}\psi_\text{p}^\star(\boldsymbol{\nabla} - \text{i}\mathbf{A})(\psi_\text{n}^\star\psi_\text{p}) \right\}, \tag{16}$$

$$\nabla^2\psi_\text{n} = R^2(|\psi_\text{n}|^2 - 1)\psi_\text{n} + \alpha(|\psi_\text{p}|^2 - 1)\psi_\text{n}$$
$$-h_1\psi_\text{p}(\boldsymbol{\nabla} + \text{i}\mathbf{A})^2(\psi_\text{p}^\star\psi_\text{n}) - \psi_\text{n}\nabla^2\left(\frac{h_2 - h_1}{2}|\psi_\text{p}|^2 + \frac{h_3}{2\epsilon}|\psi_\text{n}|^2\right), \tag{17}$$

$$(\boldsymbol{\nabla} - \text{i}\mathbf{A})^2\psi_\text{p} = (|\psi_\text{p}|^2 - 1)\psi_\text{p} + \frac{\alpha}{\epsilon}(|\psi_\text{n}|^2 - 1)\psi_\text{p}$$
$$-\frac{h_1}{\epsilon}\psi_\text{n}(\boldsymbol{\nabla} - \text{i}\mathbf{A})^2(\psi_\text{n}^\star\psi_\text{p}) - \psi_\text{p}\nabla^2\left(\frac{h_2 - h_1}{2\epsilon}|\psi_\text{n}|^2 + \frac{h_3}{2}|\psi_\text{p}|^2\right). \tag{18}$$

However, the appropriate boundary conditions for these two experiments are different, and also depend on the particular size and shape chosen for the domain. Without loss of generality, we will assume from here on that the magnetic field is oriented in the *z*-direction, and that all variables are independent of *z*; so our domain will be some region within the *xy*-plane. We solve a discretized version of Equations (16)–(18), which are obtained by minimizing a discrete approximation to the free energy on a regular grid in *x* and *y*. The gauge field is included via a Peierls substitution, in order to maintain gauge invariance. The equations are solved using a simple relaxation method, and the grid resolution is repeatedly refined until a sufficient level of accuracy has been obtained. Additional details on the numerical algorithm can be found in Appendix A.

In the present study, we are not interested in the effect of physical boundaries on the phase diagram, and so we would ideally use an infinite domain, but for numerical calculations the domain must of course be finite. Moreover, we cannot use periodic boundary conditions, because in the presence of fluxtubes neither $\mathbf{A}$ nor $\psi_\text{p}$ is spatially periodic. Instead, we must use quasi-periodic boundary conditions [53], which involves specifying not only the size of the domain, $L_x \times L_y$, say, but also the number, $N$, of magnetic flux quanta within the domain. Working in the symmetric gauge, the quasi-periodic boundary conditions for our dimensionless variables take the form

$$\mathbf{A}(\mathbf{x} + \mathbf{L}) = \mathbf{A}(\mathbf{x}) + \frac{N\pi}{L_x L_y}\mathbf{e}_z \times \mathbf{L}, \tag{19}$$

$$\psi_\text{p}(\mathbf{x} + \mathbf{L}) = \psi_\text{p}(\mathbf{x})\exp\left(\text{i}\frac{N\pi}{L_x L_y}\mathbf{e}_z \times \mathbf{L} \cdot \mathbf{x}\right), \tag{20}$$

$$\psi_\text{n}(\mathbf{x} + \mathbf{L}) = \psi_\text{n}(\mathbf{x}), \tag{21}$$

where $\mathbf{L}$ represents either of the translational symmetries $(L_x, 0)$ or $(0, L_y)$. The effect of these boundary conditions is to impose a total magnetic flux of $2\pi N$ within our domain (in our dimensionless units, the quantum of magnetic flux is $2\pi$), and thus a mean magnetic flux of $\overline{B} = 2\pi N/(L_x L_y)$. By solving Equations (16)–(18) subject to these boundary conditions, in domains of varying size, we can thus obtain the Helmholtz free energy, $\mathcal{F}$, as a

function of the mean magnetic flux. We note that the choice of domain aspect ratio affects the configuration of any fluxtube array that forms. In particular, we can impose either a square or a hexagonal lattice symmetry by using the following domain shapes:

- for a square lattice, we take $N = 1$ and $L_x/L_y = 1$;
- for a hexagonal lattice, we take $N = 2$ and $L_x/L_y = \sqrt{3}$.

In order to directly compare these two cases, we calculate the Helmholtz free energy per magnetic flux quantum per unit length:

$$\mathcal{F} \equiv \frac{1}{N} \int_{x=0}^{L_x} \int_{y=0}^{L_y} F \, dx \, dy \,. \tag{22}$$

As an example, in Figure 2 we plot $\mathcal{F}$ as a function of the area per magnetic flux quantum,

$$a \equiv \frac{L_x L_y}{N} = \frac{2\pi}{\overline{B}}, \tag{23}$$

for one particular set of parameters, whose motivation is explained later, in Section 5. Note that in the case of a fluxtube lattice, $a$ corresponds to the area of a single Wigner–Seitz cell. This plot was produced by finding the minimum value of $\mathcal{F}$ for both square and hexagonal lattices for domains of various sizes. We also plot the energy in the non-superconducting state, which has $\psi_\mathrm{p} = 0$ and a uniform magnetic field $\mathbf{B} = (0, 0, 2\pi/a)$, and is known analytically (see Section 4.1).

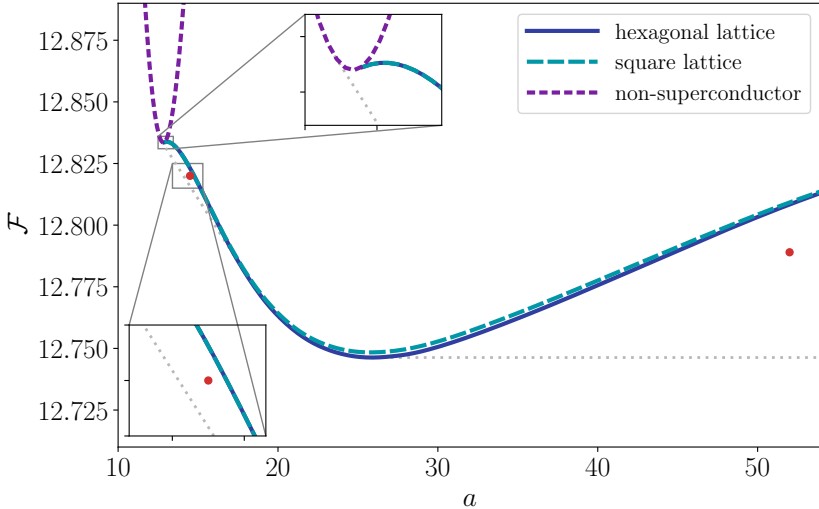

**Figure 2.** The Helmholtz free energy per flux quantum per unit length, $\mathcal{F}$, as a function of the area per magnetic flux quantum, $a$, with the dimensionless parameters $\kappa = 1.444$, $R = 0.371$, $\epsilon = 0.097$, $\alpha = 0$, $h_1 = 0.102$, $h_2 = 0.387$, and $h_3 = 0.263$. The energy in both the square (long-dashed, cyan) and hexagonal (solid, blue) lattice states matches smoothly onto the energy of the non-superconducting state (short-dashed, purple) at $a \simeq 12.9$ (region enlarged in the upper inset), and both converge to the same finite value as $a \to \infty$. The dotted gray lines indicate the lower convex envelope, plotted separately in Figure 3, which is the true ground state in an unbounded domain. The two red dots indicate the values for the two simulations in Figure 4. We show an enlarged view of the left point in the lower inset.

However, as discussed in the previous section, in some cases the true ground state might be inhomogeneous, if this allows the average energy to be lower than that of any homogeneous state. Suppose, for example, that a fraction, $\phi$, of the total magnetic flux is

contained in regions with $a = a_1$ and $\mathcal{F} = \mathcal{F}_1$, while the rest is in regions with $a = a_2$ and $\mathcal{F} = \mathcal{F}_2$. In that case, the overall values of $a$ and $\mathcal{F}$ are given by the lever rule:

$$\bar{a} = \phi a_1 + (1 - \phi)a_2, \quad \text{and} \quad \overline{\mathcal{F}} = \phi \mathcal{F}_1 + (1 - \phi)\mathcal{F}_2. \tag{24}$$

In this way, an energy $\overline{\mathcal{F}}$ that is lower than $\mathcal{F}(\bar{a})$ can be achieved in any range of $a$ for which the function $\mathcal{F}(a)$ is not convex. In fact, the true ground state in an unbounded domain is given by the lower convex envelope of all the homogeneous states, which is indicated by the dotted gray lines in Figure 2. We will use the notation $\mathcal{F}_g(a)$ to refer to the true ground-state energy as a function of the area $a$. For this particular case, there are four distinct behaviors seen across the full range of $a$:

- for $0 < a \lesssim 12.7$ the ground state is non-superconducting;
- for $12.7 \lesssim a \lesssim 18.5$ the ground state is a mixture of non-superconductor and a hexagonal fluxtube lattice;
- for $18.5 \lesssim a \lesssim 26$ the ground state is a hexagonal fluxtube lattice;
- for $a \gtrsim 26$ the ground state is a mixture of a hexagonal fluxtube lattice and the Meissner state.

Once the function $\mathcal{F}_g(a)$ is known, it is straightforward to also determine the minimum Gibbs energy as a function of $\mathbf{H}$. In fact, the mean Gibbs energy density is

$$\overline{G} = \mathcal{G}/a = \frac{\mathcal{F}_g(a)}{a} - \frac{4\pi\kappa^2}{a}|\mathbf{H}|, \tag{25}$$

and the minimum of $\overline{G}$ over all $a$ can be interpreted graphically from the plots in Figure 2. Since $\mathcal{F}_g(a)$ is a convex and monotonically decreasing function, each point on the curve $\mathcal{F}_g(a)$ corresponds to a ground state with energy $\overline{G} = \mathcal{F}'_g(a)$, and the corresponding value of $|\mathbf{H}|$ can be found by extrapolating the tangent line up to the $\mathcal{F}$-axis, as shown in Figure 3. The two ranges of $a$ for which the function $\mathcal{F}_g(a)$ is linear give rise to two critical values of $|\mathbf{H}|$ (i.e., $H_{c1'}$ and $H_{c2'}$) at which the ground state changes discontinuously. At these values there are first-order transitions between a hexagonal fluxtube lattice with finite mean field and either the Meissner state (at $H_{c1'}$) or the non-superconducting state (at $H_{c2'}$).

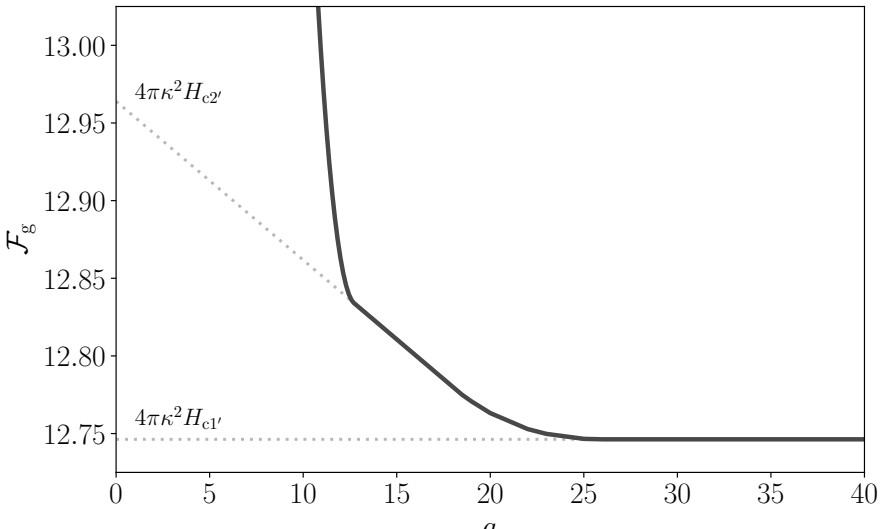

**Figure 3.** The ground-state Helmholtz free energy per unit length, $\mathcal{F}_g(a)$, can be used to infer the minimum Gibbs free-energy density, $\overline{G}$, as a function of the external field, $\mathbf{H}$. The tangent to each point on the curve $\mathcal{F}_g(a)$ has slope $\overline{G}$, and intersects the vertical axis at the point $\mathcal{F} = 4\pi\kappa^2|\mathbf{H}|$. The dotted lines indicate the transitions at $H_{c1'}$ and $H_{c2'}$, both of which are first-order in this example. We have used the same dimensionless parameters as for Figure 2.

We emphasize that the function $\mathcal{F}_{\mathrm{g}}$ represents the minimum free energy only for a hypothetical unbounded domain, free from any geometrical constraints. In any simulation with a finite domain size, the free energy in the ground state will generally exceed this value. Nevertheless, by using a large enough computational domain, and choosing values of $a$ within the appropriate ranges, we can obtain examples of the inhomogeneous ground states described above. Figure 4 shows two such examples, with $a = 14.5$ and $a = 52$. These values of $a$ were chosen so that in each case approximately half of the domain contains a hexagonal lattice. As shown in Figure 2, in each case the free energy is lower than that of a pure lattice, but still significantly higher than for the true ground state in an unbounded domain.

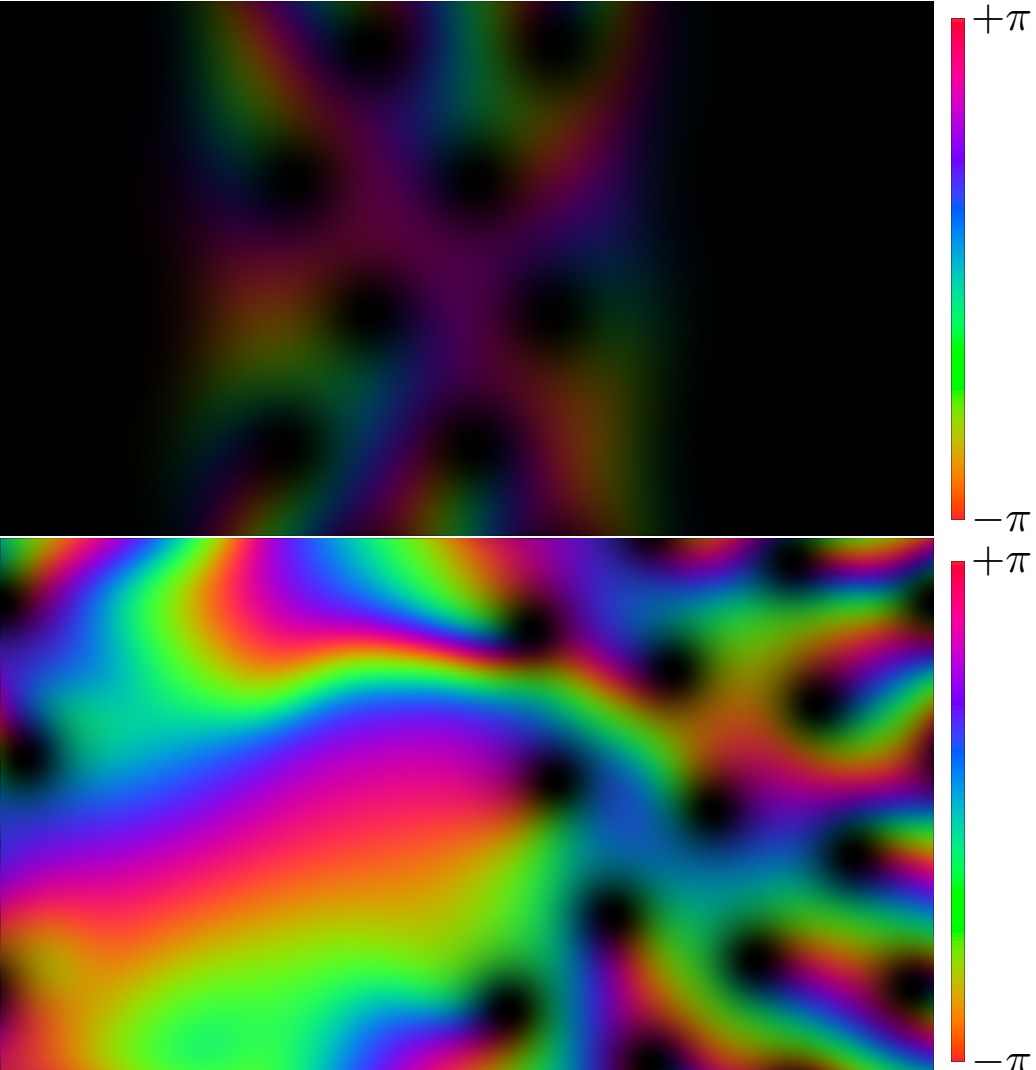

**Figure 4.** Inhomogeneous ground states for the parameters indicated by red dots in Figure 2. The brightness and hue indicate the density and phase of the proton order parameter, $\psi_{\mathrm{p}}$, respectively. As indicated by the color bar, the phase winds by $2\pi$ around each fluxtube. The first panel shows a case with $N = 24$ magnetic flux quanta, and a (dimensionless) area of $aN = 14.5 \times 24$, corresponding to a mixture of non-superconducting protons and hexagonal fluxtube lattice. Approximately 2/3 of the magnetic flux is contained in the non-superconducting domain, visible as dark bands on both sides of the image, and hence only 8 fluxtubes are visible. The low brightness of the lattice domain indicates the low density of the proton condensate there, i.e., $|\psi_{\mathrm{p}}|^2 \ll 1$. The second panel shows a case with $N = 14$ magnetic flux quanta, and a (dimensionless) area of $aN = 52 \times 14$; this is a mixture of Meissner state and hexagonal fluxtube lattice. In both cases the aspect ratio is $\sqrt{3}$, which means that a pure hexagonal lattice is a possible state of the system, but is not the ground state.

### 3.4. Phase Diagrams

Using the procedure described above, we can determine the superconducting phase transitions that occur when $\overline{B}$ or $|\mathbf{H}|$ is used as the control parameter. The results shown in Figure 2 demonstrate that these transitions can be qualitatively different from those seen in a single-component Ginzburg–Landau superconductor. In order to directly compare the resulting superconducting phases with those shown previously in Figure 1, we produce phase diagrams in which $\kappa$ is used as the independent parameter and all other parameters are fixed to the same values used in Figure 2. The result is shown in Figure 5. For $\kappa \lesssim 1.42$ we have type-I superconductivity, and for $\kappa \gtrsim 5.36$ we have type-II superconductivity. However, within the range $1.42 \lesssim \kappa \lesssim 5.36$ we have type-1.5 superconductivity: in the experiment with $|\mathbf{H}|$, the transition from the Meissner to the fluxtube state at $H_{c1'}$ is first order, because the lattice first appears with a finite separation between fluxtubes; in the experiment with $\overline{B}$, there is a critical value $\overline{B} = B_{c1}$ below which bundles of fluxtubes form alongside flux-free Meissner regions. This behavior results from the fact that fluxtubes are mutually attractive at large separation distances, and mutually repulsive at small separation distances, i.e., they have a preferred separation, as indicated by the minimum in the curve $\mathcal{F}(a)$ seen in Figure 2.

Over the narrow range $1.42 \lesssim \kappa \lesssim 1.58$ an additional feature is present: in the experiment with $|\mathbf{H}|$, the transition from the fluxtube state to the non-superconducting state is first order at $H_{c2'}$, as the proton condensate density vanishes discontinuously; in the experiment with $\overline{B}$, there is a critical value $\overline{B} = B_{c2}$ above which bundles of fluxtubes form alongside non-superconducting regions. This behavior can also be understood intuitively in terms of interactions between neighboring fluxtubes: for this range of parameters, the tendency towards a preferred separation distance is so strong, and the dip in the $\mathcal{F}(a)$ curve so pronounced, that it becomes energetically favorable to form a mixture of fluxtubes and non-superconducting regions, rather than a periodic lattice with the "wrong" separation distance.

The phase diagrams shown in Figure 5 resemble those hypothesized by Haber and Schmitt [8], and demonstrate that type-1.5 superconductivity can arise over a significant range of the Ginzburg–Landau parameter $\kappa$, when coupling between the proton and neutron condensates is present. However, to determine whether the behavior seen for this particular choice of coupling parameters is generic, we need to understand these phase transitions in more detail. In the following section we therefore seek analytical expressions for the locations of the various transitions in phase space, in order to determine which couplings between the two condensates give rise to inhomogeneous ground states.

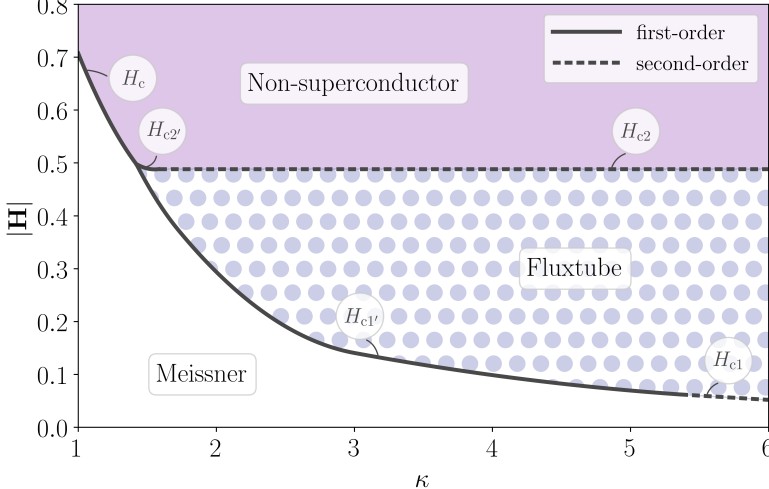

**Figure 5.** *Cont.*

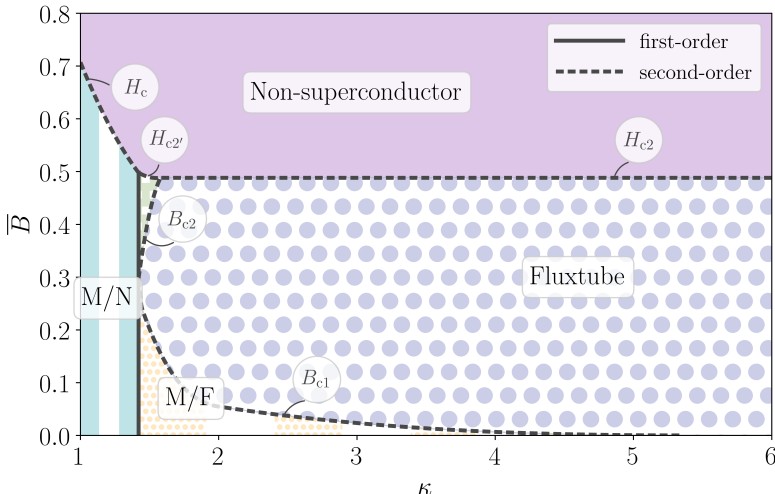

**Figure 5.** Phase diagrams for the two-component Ginzburg–Landau superconductor, for different values of the Ginzburg–Landau parameter, $\kappa$, and other parameters fixed to the same values as in Figure 2. Both figures are plotted in the same style as Figure 1. Shading of the respective regions is indicative of the magnetic flux distribution. In addition to the phases observed for the single-component case (we label the intermediate type-I phase as M/N), we also obtain inhomogeneous regimes where Meissner and fluxtube regions (M/F) as well as fluxtube and non-superconducting regions alternate. We refrain from labeling the latter to avoid overcrowding the plot. These are associated with the appearance of the critical fields $H_{c1'}$ and $H_{c2'}$, respectively.

## 4. Phase Transitions with H

In this section we will analyze in detail the phase transitions in the experiment involving the external magnetic field, **H**. As is clear from the previous section, the existence of first-order transitions at the lower and upper critical fields, $H_{c1'}$ and $H_{c2'}$, results from the non-convexity of the free energy $\mathcal{F}(a)$ in the pure lattice state. To describe these phase transitions in general requires quite detailed knowledge of the function $\mathcal{F}_g(a)$, which can only be determined with a 2D numerical model. However, some important features of the superconducting phase diagram can be determined either analytically, or from knowledge of the structure of a single fluxtube. This allows the phase diagram to be constructed more efficiently, because it reduces reliance on the 2D code, and it provides some physical insight into the origins of these first-order phase transitions. In the following sections, we describe those features of the function $\mathcal{F}(a)$ that can be determined analytically or semi-analytically.

### 4.1. The Critical Field, $H_c$

The two simplest solutions of the Euler–Lagrange Equations (16)–(18) are the Meissner (flux-free) state, which has $|\psi_p| = |\psi_n| = 1$ and $\mathbf{B} = \mathbf{0}$, and the non-superconducting state, which has $|\psi_p| = 0$, $|\psi_n| = \sqrt{1 + \alpha/R^2}$ and a uniform magnetic flux density $\mathbf{B}$ with $|\mathbf{B}| = 2\pi/a$. As discussed earlier, these two states can only be realized if the mutual repulsion/attraction $g_{pn}$, and thus $\alpha$, between the two condensates is sufficiently weak. Expressed in terms of the dimensionless parameter $\alpha$, this implies the conditions $\alpha^2 < R^2 \epsilon$ and $\alpha > -R^2$. We will assume from here on that both of these are satisfied.

To determine the thermodynamical critical field, $H_c$, we equate the energy of the Meissner state with that of the non-superconducting state. By construction, the free-energy density (12) in the Meissner state is $F = 0$, and, hence, its mean Gibbs energy is $\overline{G} = 0$. For a type-I system, the transition to the non-superconducting state therefore occurs when the corresponding energy density $\overline{G}$ becomes negative.

The purple, short-dashed curve in Figure 2 represents the free energy per unit length per quantum of flux in the non-superconducting state. From Equation (12), we obtain

$$\mathcal{F} = \frac{a}{2}\left(1 - \frac{\alpha^2}{\epsilon R^2}\right) + \frac{(2\pi\kappa)^2}{a} \, . \tag{26}$$

Substituting into Equation (25), we then find that the minimum mean Gibbs energy density, $\overline{G}$, is achieved for $a = 2\pi/|\mathbf{H}|$, as expected, and that this minimum is

$$\overline{G} = \frac{1}{2}\left(1 - \frac{\alpha^2}{\epsilon R^2}\right) - \kappa^2|\mathbf{H}|^2 \, . \tag{27}$$

Thus, assuming that we have a type-I superconductor, there is a first-order transition between the Meissner and the non-superconducting state at

$$|\mathbf{H}| = H_c \equiv \frac{1}{\sqrt{2}\kappa}\sqrt{1 - \frac{\alpha^2}{\epsilon R^2}} \, . \tag{28}$$

This defines the critical field, $H_c$, in agreement with previous studies [8,37].

*4.2. The Lower Critical Field, $H_{c1}$ vs. $H_{c1'}$*

As shown in Section 3.3, the lower critical field is a first-order phase transition if the Helmholtz free energy per unit length per flux quantum $\mathcal{F}(a)$ in the pure lattice state has a minimum at some finite value of $a$, since then a fluxtube lattice forms with finite separation. If this minimum is $\mathcal{F}_{\min}$, say, then we have $H_{c1'} = \mathcal{F}_{\min}/(4\pi\kappa^2)$. If, on the other hand, $\mathcal{F}$ is a monotonically decreasing function of $a$ in the lattice state, then there is a second-order transition at $H_{c1} = \mathcal{F}_\infty/(4\pi\kappa^2)$, as in the case of a single-component superconductor, where

$$\mathcal{F}_\infty = \lim_{a\to\infty}\mathcal{F}(a) \, . \tag{29}$$

In this limit, interactions between the fluxtubes vanish, and $\mathcal{F}_\infty$ is equivalent to the energy per unit length of a single fluxtube in an infinite domain. This energy can be computed efficiently by using polar coordinates $r, \theta$ centered on the fluxtube core, and seeking solutions of Equations (16)–(18) in the form [7]

$$\psi_p = f(r)\,e^{i\theta}, \quad \psi_n = g(r), \quad \mathbf{A} = A_\theta(r)\,\mathbf{e}_\theta \, . \tag{30}$$

This ansatz assumes that the fluxtube carries a single quantum of magnetic flux, and that there is no corresponding phase defect in the neutron condensate. It further results in a system of ordinary differential equations that can be solved numerically, yielding the value of $\mathcal{F}_\infty$ to high accuracy. However, to determine whether the lower transition is second-order or first-order, we need to know whether the function $\mathcal{F}(a)$ tends to $\mathcal{F}_\infty$ from above or from below, which is equivalent to asking whether the long-range interaction between fluxtubes is repulsive or attractive. This can be derived rigorously using a method introduced by Kramer [54], which we describe in detail in Appendix B. In fact, the main result can be obtained heuristically by considering the perturbations produced by a fluxtube in the far-field, i.e., at a large distance from its core. It is convenient to work with the real variables

$$f \equiv |\psi_p|, \quad g \equiv |\psi_n|, \quad \chi \equiv \arg\psi_n, \quad \mathbf{V} \equiv \boldsymbol{\nabla}(\arg\psi_p) - \mathbf{A} \, . \tag{31}$$

The gauge invariance of the free energy density (12) guarantees that it can be rewritten in terms of these variables without loss of generality; for the full expression see Equation (A6). The corresponding Euler–Lagrange equations are then

$$0 = f(f^2 - 1) + \frac{\alpha}{\epsilon} f(g^2 - 1) - \nabla^2 f + f|\mathbf{V}|^2$$

$$+ \frac{h_1}{\epsilon}[f|\nabla g|^2 + fg^2|\mathbf{V} - \nabla\chi|^2 - \nabla \cdot (g^2 \nabla f)] - \frac{h_2}{2\epsilon} f\nabla^2(g^2) - \frac{h_3}{2} f\nabla^2(f^2), \tag{32}$$

$$0 = \frac{R^2}{\epsilon} g(g^2 - 1) + \frac{\alpha}{\epsilon} g(f^2 - 1) - \frac{1}{\epsilon}\nabla^2 g + \frac{1}{\epsilon} g|\nabla\chi|^2$$

$$+ \frac{h_1}{\epsilon}[g|\nabla f|^2 + gf^2|\mathbf{V} - \nabla\chi|^2 - \nabla \cdot (f^2 \nabla g)] - \frac{h_2}{2\epsilon^2} g\nabla^2(f^2) - \frac{h_3}{2\epsilon^2} g\nabla^2(g^2), \tag{33}$$

$$0 = \nabla \cdot [g^2 \nabla\chi + h_1 f^2 g^2 (\nabla\chi - \mathbf{V})], \tag{34}$$

$$0 = f^2 \mathbf{V} + \frac{h_1}{\epsilon} f^2 g^2 (\mathbf{V} - \nabla\chi) + \kappa^2 \nabla \times (\nabla \times \mathbf{V}). \tag{35}$$

The far-field structure of the fluxtube can be determined by linearizing these about the uniform state with $f = g = 1$, and $\mathbf{V} = \nabla\chi = \mathbf{0}$. This leads to the following system:

$$\left(1 + \frac{h_1}{\epsilon} + h_3\right)\nabla^2 \delta f + \frac{h_2}{\epsilon}\nabla^2 \delta g = 2\delta f + \frac{2\alpha}{\epsilon}\delta g, \tag{36}$$

$$\left(1 + h_1 + \frac{h_3}{\epsilon}\right)\nabla^2 \delta g + h_2 \nabla^2 \delta f = 2R^2 \delta g + 2\alpha\delta f, \tag{37}$$

$$\kappa^2 \nabla \times (\nabla \times \delta\mathbf{V}) = \frac{h_1}{\epsilon}\nabla\delta\chi - \left(1 + \frac{h_1}{\epsilon}\right)\delta\mathbf{V}, \tag{38}$$

$$(1 + h_1)\nabla^2 \delta\chi = h_1 \nabla \cdot \delta\mathbf{V}. \tag{39}$$

where $\delta f$, $\delta g$, $\delta\mathbf{V}$, and $\delta\chi$ denote the linear perturbations. In the case of a single fluxtube, we are interested in the solution that is axisymmetric and decays at large distance from the origin. We deduce from Equations (38) and (39) that this solution has $\delta\chi = 0$ and

$$\delta\mathbf{V} = V_0 K_1\left(\frac{r}{\lambda_\star}\right)\mathbf{e}_\theta, \tag{40}$$

for some constant coefficient $V_0$, where $K_1$ is a modified Bessel function and $\lambda_\star$ is the effective London length,

$$\lambda_\star = \kappa\left(1 + \frac{h_1}{\epsilon}\right)^{-1/2}. \tag{41}$$

Note that, compared to the London length $\lambda$ in the absence of coupling, $\lambda_\star$ is made smaller by the parameter $h_1$, because the effective mass of the protons is made smaller by the entrainment of neutrons [39]. From Equations (36) and (37) we find that, owing to the coupling between the condensates, the fluxtube in the far-field has a double-coherence-length structure, i.e.,

$$\delta f = f_1 K_0\left(\sqrt{2}r/\xi_1\right) + f_2 K_0\left(\sqrt{2}r/\xi_2\right), \tag{42}$$

$$\delta g = g_1 K_0\left(\sqrt{2}r/\xi_1\right) + g_2 K_0\left(\sqrt{2}r/\xi_2\right), \tag{43}$$

where $K_0$ is a zeroth-order modified Bessel function. The coefficients $f_i, g_i$ and the effective coherence lengths $\tilde{\xi}_i$ satisfy the equations

$$\left(1 + \frac{h_1}{\epsilon} + h_3\right)\frac{f_i}{\tilde{\xi}_i^2} + \frac{h_2}{\epsilon}\frac{g_i}{\tilde{\xi}_i^2} = f_i + \frac{\alpha}{\epsilon}g_i, \tag{44}$$

$$\left(1 + h_1 + \frac{h_3}{\epsilon}\right)\frac{g_i}{\tilde{\xi}_i^2} + h_2\frac{f_i}{\tilde{\xi}_i^2} = R^2 g_i + \alpha f_i. \tag{45}$$

This is reminiscent of the fluxtube structure found in two-component superconductors, although in that case the two coherence lengths arise because a fluxtube is a phase defect in *both* condensates [55]. In our model, fluxtubes are phase defects in the proton condensate only, but the coupling coefficients $\alpha$ and $h_2$ ultimately achieve a similar effect. In some parameter regimes (as we have indeed seen in Section 3.4), we might therefore expect our model to exhibit type-1.5 superconductivity, which is common in two-component superconductors, and generally occurs when the London length lies between the two coherence lengths [13]. The physical reason is that the electromagnetic interaction between fluxtubes, which decays on the London length, is generally repulsive, whereas the density interactions are generally attractive. Thus type-1.5 superconductivity arises because fluxtubes are mutually attractive at large separation distances, but become mutually repulsive at shorter separations, producing bundles of fluxtubes with a preferred density, as indicated by the minimum in the $\mathcal{F}(a)$ curve.

As mentioned earlier, this heuristic argument can be put on a more rigorous basis by calculating the long-range interaction energy between fluxtubes in a lattice configuration, using the method first described by Kramer [54]. A similar method was used by Haber and Schmitt [8] to demonstrate the existence of type-1.5 superconductivity resulting from density and derivative couplings. As we evaluate the interaction energy for a more general case than [8], and thus obtain a different result, we present our analysis in full in Appendix B, although the steps closely follow those of Kramer [54] for a single-component superconductor. Our final result for the interaction energy is

$$\mathcal{F} - \mathcal{F}_\infty \simeq 2\pi\kappa^2 V_0^2 \sum_{i\neq 0} K_0(|\mathbf{x}_i|/\lambda_\star) - 2\pi\sum_{j=1,2}\left[f_j^2 + 2\frac{\alpha}{\epsilon}f_j g_j + \frac{R^2}{\epsilon}g_j^2\right]\tilde{\xi}_j^2 \sum_{i\neq 0} K_0\left(\sqrt{2}|\mathbf{x}_i|/\tilde{\xi}_j\right), \tag{46}$$

where $\mathbf{x}_i$ is the location of the $i$-th lattice point, assuming that $\mathbf{x}_0 = \mathbf{0}$. In the absence of any coupling between the two fluids ($\alpha = h_1 = h_2 = h_3 = 0$) this reduces exactly to the result of Kramer [54].

In principle, we can now use this result to estimate the free energy of a particular lattice state using just the values of the coefficients $f_i, g_i, V_0$, which can themselves be inferred from the nonlinear solution for a single fluxtube. However, this result is only strictly valid if the fluxtubes are very widely separated, and it becomes inaccurate once the fluxtubes are close enough to interact nonlinearly. Nevertheless, we can deduce that, in the asymptotic limit $a \to \infty$,

$$\mathcal{F} - \mathcal{F}_\infty \propto \kappa^2 V_0^2\sqrt{\frac{\lambda_\star}{d}}\, e^{-d/\lambda_\star} - \left[f_+^2 + 2\frac{\alpha}{\epsilon}f_+ g_+ + \frac{R^2}{\epsilon}g_+^2\right]\tilde{\xi}_+^2\sqrt{\frac{\tilde{\xi}_+}{\sqrt{2}d}}\, e^{-\sqrt{2}d/\tilde{\xi}_+} \tag{47}$$

where $d \propto \sqrt{a}$ is the lattice constant, and $\tilde{\xi}_+$ represents the larger of the two effective coherence lengths, which in practice is always larger than both of the bare coherence lengths. If $\lambda_\star > \tilde{\xi}_+/\sqrt{2}$ then, according to Equation (47), the long-range interaction energy is positive, implying that there is a second-order transition at the lower critical field $H_{c1}$, as for a type-II superconductor. Conversely, if $\lambda_\star < \tilde{\xi}_+/\sqrt{2}$ then the interaction energy is negative, implying that the $\mathcal{F}(a)$ curve has a minimum at a finite value of $a$. In that case, there is a first-order transition at the lower critical field $H_{c1'}$, and we have a type-1.5 superconductor, which confirms the heuristic argument given above. In summary, we can

identify the precise value of $\kappa$ at which $H_{c1} = H_{c1'}$ (and $B_{c1} = 0$) by equating $\lambda_\star$, which is given by Equation (41), with $\xi_+/\sqrt{2}$, which is independent of $\kappa$. In the case shown in Figure 5, this yields the value $\kappa \simeq 5.36$, in agreement with our numerical results.

*4.3. The Upper Critical Field, $H_{c2}$ vs. $H_{c2'}$*

As shown in Section 3.3, if the free energy $\mathcal{F}(a)$ is convex (and monotonically decreasing), then the transition between the fluxtube lattice state and the non-superconducting state is second-order. This means that the order parameter $\psi_p$ vanishes smoothly at the transition point, $H_{c2}$, which can thus be determined analytically by considering linear perturbations to the non-superconducting state. Working in the symmetric gauge, the non-superconducting state is given by $|\psi_p| = 0$, $|\psi_n| = \sqrt{1 + \alpha/R^2}$, and $\mathbf{A} = \frac{1}{2}\overline{B}\mathbf{e}_z \times \mathbf{x}$, where $\overline{B} = 2\pi/a$ is the (uniform) mean magnetic flux. From the linearized version of Equation (18), we find that the perturbation $\delta\psi_p$ must satisfy the equation

$$\left[1 + \frac{h_1}{\epsilon}\left(1 + \frac{\alpha}{R^2}\right)\right]\left(\nabla - \tfrac{1}{2}i\overline{B}\mathbf{e}_z \times \mathbf{x}\right)^2 \delta\psi_p = -\left(1 - \frac{\alpha^2}{\epsilon R^2}\right)\delta\psi_p\,. \tag{48}$$

As known from single-component systems, e.g., [52], bounded solutions of this equation first appear when

$$\overline{B} = H_{c2} \equiv \frac{1 - \frac{\alpha^2}{\epsilon R^2}}{1 + \frac{h_1}{\epsilon}\left(1 + \frac{\alpha}{R^2}\right)}\,. \tag{49}$$

Therefore, the solid (blue) and long-dashed (cyan) lines in Figure 2, as highlighted in the upper inset, meet the short-dashed (purple) line at the point where $a = 2\pi/H_{c2}$. However, if the function $\mathcal{F}(a)$ is not convex at this point then the upper transition will occur not at $|\mathbf{H}| = H_{c2}$, but at a higher value $|\mathbf{H}| = H_{c2'}$, and will be first order. To determine whether the function $\mathcal{F}(a)$ is convex at this point, we can seek weakly nonlinear solutions in the vicinity of $a = 2\pi/H_{c2}$, following the method of Abrikosov [56]. We present the details of this calculation in Appendix C. The main result is that the function $\mathcal{F}(a)$ becomes non-convex when

$$\left[\frac{2}{H_{c2}} + h_3 - \left(1 - \frac{\alpha^2}{\epsilon R^2}\right)^2 \frac{1}{\kappa^2 H_{c2}^3}\right]\pi\beta\epsilon R^2 =$$

$$\sum_i \exp(-\tfrac{1}{2}H_{c2}|\mathbf{x}_i|^2)\frac{\left[(h_2 - h_1)\frac{1}{2}H_{c2}|\mathbf{x}_i|^2 + h_1 + \alpha\frac{1}{H_{c2}}\right]^2}{\left(\frac{1}{R^2+\alpha} + \frac{h_3}{\epsilon R^2}\right)\frac{1}{2}H_{c2}|\mathbf{x}_i|^2 + \frac{1}{H_{c2}}}\,, \tag{50}$$

where $\mathbf{x}_i$ is the location of the $i$-th lattice point, as in Section 4.2, and $\beta$ is the kurtosis of the proton order parameter,

$$\beta \equiv \frac{\overline{|\psi_p|^4}}{(\overline{|\psi_p|^2})^2} = \sum_i \exp(-\tfrac{1}{2}H_{c2}|\mathbf{x}_i|^2)\,. \tag{51}$$

Equation (50) is valid for both hexagonal and square lattices of singly-charged fluxtubes. (Singly-charged here means that each fluxtube carries a single quantum of magnetic flux.) Although there are cases in which other fluxtube configurations have a lower free energy (see Appendix C), the point of transition between $H_{c2}$ and $H_{c2'}$ is always determined by the singly-charged hexagonal lattice, for which $\beta \simeq 1.1596$ [57]. Thus from Equation (50) we can determine the value of $\kappa$ at which $H_{c2}$, $H_{c2'}$ and $B_{c2}$ all coincide. In the case shown in Figure 5, this yields the value $\kappa \simeq 1.58$, in agreement with our numerical results.

## 5. Discussion

Combining analytical and numerical techniques, we have determined the ground state for a mixture of proton and neutron superfluid condensates in the neutron star core,

in the presence of magnetic flux. The condensates are coupled by density and density-gradient interactions, and by mutual entrainment. Our model extends the phenomenological Ginzburg–Landau framework used in previous works to consistently satisfy Galilean invariance on small scales. In addition to the three homogeneous phases (Meissner, fluxtube lattice, and non-superconducting) that are familiar from single-component superconductors, we have shown that inhomogeneous mixtures of any two of these phases can occur in the ground state, as a result of the coupling between the condensates. The novel mixed states occur because the fluxtubes perturb the neutron condensate density, which can reduce the overall free energy of the coupled system. As described in Section 4.2, coupling between the two condensates always increases the effective coherence length of fluxtubes, and hence under a range of conditions the fluxtubes have a preferred separation distance. Under these conditions, and for relatively weak mean field values ($\overline{B} < B_{c1}$), we have type-1.5 superconductivity, in which the fluxtubes form bundles with local hexagonal symmetry, with the rest of the domain in a flux-free Meissner state. In cases where the tendency towards fluxtube bundling is sufficiently strong, there may also be a range of mean field values ($B_{c2} < \overline{B} < H_{c2'}$) for which there is a mixture of fluxtube and non-superconducting regions. This occurs if it is energetically favorable to confine the proton condensate within part of the domain, forming a lattice with the preferred separation, with the remaining magnetic flux concentrated into non-superconducting regions. In all cases, we find that fluxtubes preferentially adopt a singly-charged, hexagonal pattern, even in cases where a periodic lattice is not the ground state.

To relate our results to neutron stars, we must provide estimates for all of the dimensionless parameters in our model. The results presented in Section 3 assumed the following dimensional values: $n_p = 0.0251\,1/\text{fm}^3$, $n_n = 0.258\,1/\text{fm}^3$, $\xi_p = 31.4\,\text{fm}$, $\xi_n = 84.5\,\text{fm}$, $h_1 = 85\,\text{MeV}\,\text{fm}^5$, $h_2 = 323\,\text{MeV}\,\text{fm}^5$, and $h_3 = 219\,\text{MeV}\,\text{fm}^5$. This means that the magnetic field in Figure 5 is measured in units of $\hbar c/(2e\xi_p^2) \simeq 3 \times 10^{15}\,\text{G}$. These parameter values are reasonable for the core of a neutron star, reflecting realistic nuclear matter equations of state [58] and energy gaps [47,59]. In particular, we have determined $h_1$ following Chamel and Haensel [31] for the NRAPR equation of state [60], and we have chosen $h_2$ and $h_3$ to be of similar magnitude to $h_1$. Our analytical results, presented in the previous section, indicate that the different phases seen in Figure 5 are generic, rather than an artefact of our particular parameter choice. For plausible choices of the coupling parameters $\alpha$ and $h_2$, we always find type-1.5 superconductivity for a significant range of $\kappa$ values. This suggests that much of the outer core is a type-1.5 superconductor, rather than type-II as is commonly assumed. However, the other notable feature of Figure 5—the existence of a mixed phase for $B_{c2} < \overline{B} < H_{c2'}$—is present only for sufficiently large values of the parameters $h_1$ or $h_2$, which may explain why a phase of this kind was not seen in the results of Kobyakov [37]. In any event, such a phase could only be found in stars with a mean flux of order $H_{c2} \sim 10^{15}\,\text{G}$.

Since the pioneering work of Baym et al. [1], it has generally been accepted that the outer core of a neutron star is a type-II superconductor, in which fluxtubes are stable but mutually repulsive on all scales. In that case, fluxtubes are expected to form a lattice arrangement that expands over time, eventually leading to a flux-free state (albeit on a very long timescale). Our results suggest that fluxtubes have a preferred separation distance throughout a large part of the core, as a result of entrainment between the protons and neutrons, and will therefore form "hexagonal bundles" that can persist indefinitely.

Furthermore, previous works have often assumed, without providing justification, that the transition between the type-II outer core and type-I inner core occurs where the ratio of the effective London length, $\lambda_\star$, to the proton coherence length, $\xi_p$, takes the value $1/\sqrt{2}$, e.g., [39,47,61]. In terms of our dimensionless variables, this condition can be expressed as

$$\kappa = \sqrt{\frac{1}{2} + \frac{h_1}{2\epsilon}}. \tag{52}$$

However, our results demonstrate that the transition to the type-I inner core generally occurs where the critical fields $H_c$ and $H_{c2'}$ are equal. Although there is no analytical

expression for the latter, it is generally quite close to $H_{c2}$, and so we have the approximate condition $H_c \simeq H_{c2}$. In terms of our dimensionless variables, this condition becomes

$$\kappa \simeq \left[ 1 + \frac{h_1}{\epsilon} \left( 1 + \frac{\alpha}{R^2} \right) \right] \left( 2 - \frac{2\alpha^2}{\epsilon R^2} \right)^{-1/2}. \tag{53}$$

An analogous result was obtained by Kobyakov [37]. In a neutron star we typically expect $h_1/\epsilon$ to be of order unity and so, even if we neglect the density coupling parameter $\alpha$, the type-I inner core will be significantly larger than predicted by Equation (52). We note that an alternative method to determine the point of transition to type-I superconductivity is to measure the "surface energy" of the interface between Meissner and non-superconducting regions [37,62,63]. The surface energy in a type-I superconductor is necessarily positive, because otherwise the interface is unstable to the formation of fluxtubes. However, this provides only a necessary condition for stability; the interface could still be unstable to nonlinear perturbations even if the surface energy is positive. Given that we have found the non-superconducting state to be metastable in some cases (i.e., unstable to nonlinear perturbations), we speculate that in those cases the boundary between the type-I and type-1.5 regimes does not occur exactly where the surface energy is zero. Further work will be required to confirm this however.

Although we have only considered the ground state for the condensates in this work, we expect the true microphysical state in the neutron star core to be qualitatively similar, provided that the temperature is well below the condensation temperature. However, the inhomogeneous ground states may be fragile, in the sense that the difference in energy density between the ground state and the fluxtube lattice state, $(\mathcal{F}(a) - \mathcal{F}_g(a))/a \sim 10^{-3}$, is small. (For the values of $n_p$ and $\xi_p$ quoted above, this corresponds to a difference in energy density of $\sim 2 \times 10^{26}$ erg/cm³.) To understand the time-dependent dynamics of the fluxtubes in detail, including effects of finite temperature, will require a more complete dynamical model than that presented here. In particular, in this work we have considered defects in the proton condensate only, i.e., we have neglected the presence of neutron vortices that will appear as a result of the superfluid's quantized rotation. This is justifiable when characterizing the system's ground state, because the fluxtubes outnumber the vortices by many orders of magnitude, but interactions between both types of defects are crucial for the dynamics of the rotation and magnetic field on larger scales. A fully dynamical description of the neutron star interior must also incorporate the (non-superfluid) electrons, whose scattering by fluxtubes is a dominant source of dissipation in the star's core [39,64,65]. Finding a consistent treatment of the electrons, whose mean free path far exceeds the typical distance between fluxtubes, is beyond the scope of the present work and is left for future study.

**Author Contributions:** Formal analysis, T.S.W. and V.G.; Investigation, T.S.W. and V.G.; Methodology, T.S.W. and V.G.; Visualization, T.S.W. and V.G.; Writing—original draft, T.S.W. and V.G.; Writing—review and editing, T.S.W. and V.G. All authors have read and agreed to the published version of the manuscript.

**Funding:** Part of TSW's time was funded by EPSRC Grant EP/R024952/1 and by STFC grant ST/W001020/1. VG acknowledges partial support from a McGill Space Institute postdoctoral fellowship and the Trottier Chair in Astrophysics and Cosmology as well as the H2020 ERC Consolidator Grant "MAGNESIA" under grant agreement No. 817661 (PI: Rea) and Spanish National Grant PGC2018-095512-BI00. This work was also partially supported by the program Unidad de Excelencia María de Maeztu CEX2020-001058-M, and by the PHAROS COST Action (No. CA16214).

**Data Availability Statement:** The data used to produce the plots in this paper are available at https://doi.org/10.25405/data.ncl.c.5935723, accessed on 8 April 2022.

**Acknowledgments:** The authors would like to thank the Institute for Nuclear Theory at the University of Washington for its kind hospitality and hosting INT Program INT-19-1a during which part of this work was carried out. We also thank Wynn Ho, John Miller, and Hayder Salman for helpful conversations related to this paper and Alexander Haber for providing feedback on our manuscript. Finally, we acknowledge the use of the following software: IPython [66], Matplotlib [67], NumPy [68–70], Pandas [71], and SciPy [72,73].

**Conflicts of Interest:** The authors declare no conflict of interest.

## Appendix A. Minimization of the Free Energy

The dimensionless free-energy density, given by Equation (12), is approximated numerically on a regular 2D grid, with intervals $\delta x$ and $\delta y$ in the $x$ and $y$ directions. The order parameters $\psi_{\rm p}$ and $\psi_{\rm n}$ are defined on the gridpoints as $\psi_{\rm p}^{i,j}$ and $\psi_{\rm n}^{i,j}$, where $i$ and $j$ denote the indices in $x$ and $y$, respectively. The vector field $\mathbf{A}$ has only two components, $(A_x, A_y)$, which are defined on the corresponding links between the gridpoints, i.e., we have $A_x^{i+1/2,j}$ and $A_y^{i,j+1/2}$. The gauge coupling between $\psi_{\rm p}$ and $\mathbf{A}$ is implemented using a standard Peierls substitution, noting that for instance

$$\left| \left( \frac{\partial}{\partial x} - \mathrm{i} A_x \right) \psi_{\rm p} \right| = \left| \frac{\partial}{\partial x} \exp\left( - \int \mathrm{i} A_x \, \mathrm{d}x \right) \psi_{\rm p} \right|$$

$$\Rightarrow \left| \left( \frac{\partial}{\partial x} - \mathrm{i} A_x \right) \psi_{\rm p} \right|^{i+1/2,j} \simeq \frac{1}{\delta x} \left| \exp\left( -\mathrm{i} A_x^{i+1/2,j} \, \delta x \right) \psi_{\rm p}^{i+1,j} - \psi_{\rm p}^{i,j} \right| .$$

By including the gauge coupling in this way, we exactly preserve the gauge symmetry

$$\psi_{\rm p}^{i,j} \to \exp(\mathrm{i}\phi^{i,j}) \psi_{\rm p}^{i,j} ,$$

$$A_x^{i+1/2,j} \to A_x^{i+1/2,j} + \frac{\phi^{i+1,j} - \phi^{i,j}}{\delta x} ,$$

$$A_y^{i,j+1/2} \to A_y^{i,j+1/2} + \frac{\phi^{i,j+1} - \phi^{i,j}}{\delta y} . \tag{A1}$$

This leads to a discrete approximation to the total free energy, $\mathcal{F}_{\rm dis}[\psi_{\rm p}^{i,j}, \psi_{\rm n}^{i,j}, A_x^{i+1/2,j}, A_y^{i,j+1/2}]$. We obtain the ground state using a simple gradient-descent method, in which the step size is made as large as possible while maintaining numerical stability. Specifically, we use the iteration scheme

$$\psi_{\rm p}^{i,j} \to \psi_{\rm p}^{i,j} - \left( \frac{N/4}{\delta y/\delta x + \delta x/\delta y} \right) \frac{\partial \mathcal{F}_{\rm dis}/\partial \psi_{\rm p}^{\star,i,j}}{1 + \frac{h_1}{\epsilon} |\psi_{\rm n}^{i,j}|^2 + h_3 |\psi_{\rm p}^{i,j}|^2} , \tag{A2}$$

$$\psi_{\rm n}^{i,j} \to \psi_{\rm n}^{i,j} - \left( \frac{N/4}{\delta y/\delta x + \delta x/\delta y} \right) \frac{\partial \mathcal{F}_{\rm dis}/\partial \psi_{\rm n}^{\star,i,j}}{1 + h_1 |\psi_{\rm p}^{i,j}|^2 + \frac{h_3}{\epsilon} |\psi_{\rm n}^{i,j}|^2} , \tag{A3}$$

$$A_x^{i+1/2,j} \to A_x^{i+1/2,j} - \left( \frac{N/4}{\delta y/\delta x + \delta x/\delta y} \right) \frac{\partial \mathcal{F}_{\rm dis}/\partial A_x^{i+1/2,j}}{2\kappa^2} , \tag{A4}$$

$$A_y^{i,j+1/2} \to A_x^{i,j+1/2} - \left( \frac{N/4}{\delta y/\delta x + \delta x/\delta y} \right) \frac{\partial \mathcal{F}_{\rm dis}/\partial A_y^{i,j+1/2}}{2\kappa^2} . \tag{A5}$$

We use an initial (dimensionless) resolution of $\delta x, \delta y \simeq 0.5$, and iterate until the change in energy drops below a threshold of $10^{-7}$. We then double the resolution in $x$ and $y$ and repeat the whole process until the value of $\mathcal{F}_{\rm dis}$ converges to at least five significant figures.

**Appendix B. Long-Range Interaction between Fluxtubes**

The dimensionless free-energy density (12), when written in terms of the real variables $f$, $g$, $\mathbf{V}$ and $\chi$ defined in Section 4.2, takes the form

$$
\begin{aligned}
F[f,g,\mathbf{V},\chi] = \ & \frac{1}{2}(f^2-1)^2 + \frac{R^2}{2\epsilon}(g^2-1)^2 + \frac{\alpha}{\epsilon}(f^2-1)(g^2-1) \\
& + |\boldsymbol{\nabla}f|^2 + f^2|\mathbf{V}|^2 + \frac{1}{\epsilon}|\boldsymbol{\nabla}g|^2 + \frac{1}{\epsilon}g^2|\boldsymbol{\nabla}\chi|^2 + \kappa^2|\boldsymbol{\nabla}\times\mathbf{V}|^2 \\
& + \frac{h_1}{\epsilon}\left[f^2|\boldsymbol{\nabla}g|^2 + g^2|\boldsymbol{\nabla}f|^2 + f^2g^2|\mathbf{V}-\boldsymbol{\nabla}\chi|^2\right] \\
& + \frac{h_2}{2\epsilon}\boldsymbol{\nabla}f^2\cdot\boldsymbol{\nabla}g^2 + \frac{h_3}{4}\left(\frac{1}{\epsilon^2}\left|\boldsymbol{\nabla}g^2\right|^2 + \left|\boldsymbol{\nabla}f^2\right|^2\right).
\end{aligned}
\tag{A6}
$$

The reason for working with these real variables will be explained shortly. Recall that $F = 0$ in the absence of fluxtubes, vortices and magnetic flux, i.e., for $f = g = 1$ and $\mathbf{V} = \boldsymbol{\nabla}\chi = \mathbf{0}$. In what follows, we will refer to this as the Meissner solution.

Our goal is to determine the free energy per unit length per Wigner–Seitz cell, $\mathcal{F}(a)$, in the asymptotic limit of a widely-spaced fluxtube lattice, $a \to \infty$. In this limit, we know that $\mathcal{F}$ converges to the value for a single fluxtube, i.e., $\mathcal{F} \to \mathcal{F}_\infty$, and we wish to estimate the "interaction energy" $\mathcal{F} - \mathcal{F}_\infty$. If its value is positive, the lower transition is of second-order, whereas a negative value implies a first-order transition. To do so, we will first generalize the method introduced by Kramer [54] to the case of a free energy in the completely general form $\mathcal{F} = \langle F[\Psi]\rangle$, where $\Psi$ represents the complete set of independent, real variables, and the angled brackets represent a domain integral. We will then apply the results to the particular case given by Equation (A6). We consider an infinite lattice of parallel fluxtubes that are widely separated, in the sense that the size of each Wigner–Seitz cell is large in comparison with both the coherence length and the London length. Any such lattice, characterized by $\Psi$, is a steady state, in that it is a solution of the Euler–Lagrange equations that are obtained from the functional derivative

$$
\frac{\delta}{\delta\Psi}\langle F\rangle = 0.
\tag{A7}
$$

We assume that each fluxtube is located far inside its Wigner–Seitz cell, which allows us to make two approximations:

1. Within each cell, the solution can be approximated as a linear perturbation to the single fluxtube solution;
2. On the boundary of the cell, the solution can be approximated as the Meissner solution plus a superposition of linear, *independent* perturbations produced by the fluxtubes.

Note that we do not assume that the fluxtube is located exactly at the center of the cell, and we will verify later that the exact location of the fluxtube within the cell does not affect our result for the interaction energy.

We will use the notation $\delta^n F[\delta\Psi;\Psi]$ to represent the $n$-th variation of $F$, i.e., the terms up to order $(\delta\Psi)^n$ in the Taylor expansion of $F[\Psi+\delta\Psi] - F[\Psi]$. If $\Psi$ is a solution of the Euler–Lagrange Equation (A7), then $\delta^1 F[\delta\Psi;\Psi]$ is an exact derivative. This follows directly from the definition of the functional derivative, which dictates that

$$
\delta^1 F = \delta\Psi \cdot \underbrace{\left(\frac{\delta}{\delta\Psi}\langle F\rangle\right)}_{\text{EL equations}} + \boldsymbol{\nabla}\cdot\mathbf{Q},
\tag{A8}
$$

for some vector field $\mathbf{Q}[\delta\Psi;\Psi]$. Using the divergence theorem, the integral of $\delta^1 F$ over any domain can thus be expressed as an integral over the boundary of that domain.

Furthermore, if $\delta\Psi$ is a solution of the *linearized* Euler–Lagrange equations (i.e., linearized about $\Psi$), then it can be shown that $\delta^2 F[\delta\Psi;\Psi]$ is also an exact derivative. In fact, we have

$$\delta^2 F = \delta\Psi \cdot \underbrace{\left( \frac{\delta}{\delta\Psi} \langle F \rangle \right)}_{\text{EL equations}} + \frac{1}{2}\,\delta\Psi \cdot \underbrace{\left( \delta^1 \left\{ \frac{\delta}{\delta\Psi} \langle F \rangle \right\} \right)}_{\substack{\text{linearized} \\ \text{EL equations}}} + \boldsymbol{\nabla} \cdot \mathbf{Q}^{(2)} , \tag{A9}$$

where the vector field $\mathbf{Q}^{(2)}[\delta\Psi;\Psi]$ is given precisely by the terms up to order $(\delta\Psi)^2$ in the Taylor expansion of $\mathbf{Q}[\delta\Psi; \Psi + \frac{1}{2}\delta\Psi]$. Once we identify the functional form of $\delta^1 F$, we deduce $\mathbf{Q}$ and thence $\mathbf{Q}^{(2)}$. We can then express the integral of $\delta^2 F$ over any domain as an integral over the boundary of that domain. The importance of this result in determining the interaction energy will become more obvious shortly. In what follows, we refer to the linear equations for $\delta\Psi$ as the Jacobi equations, by analogy with the equations defining Jacobi fields in Riemannian geometry [74].

Suppose the fluxtubes are aligned with the $z$-axis, at locations in the $xy$-plane indexed as $\mathbf{x}_i$. Without loss of generality, we will assume that $\mathbf{x}_0 = \mathbf{0}$, and that $\mathbf{x}_{-i} = -\mathbf{x}_i$. We will also label the Wigner–Seitz cells as $C_i$, such that $\mathbf{x}_i \in C_i$. This allows us to express the interaction energy as

$$\mathcal{F} - \mathcal{F}_\infty = \iint_{C_0} F^{(\text{all})}\, \mathrm{d}x\, \mathrm{d}y - \iint_{\mathbb{R}^2} F^{(0)}\, \mathrm{d}x\, \mathrm{d}y$$

$$= \iint_{C_0} \left[ F^{(\text{all})} - \sum_i F^{(i)} \right] \mathrm{d}x\, \mathrm{d}y , \tag{A10}$$

where $F^{(\text{all})}$ represents the free-energy density in the presence of the lattice, and $F^{(i)}$ represents the free-energy density in the presence of a single fluxtube at $\mathbf{x} = \mathbf{x}_i$. To obtain the last line, we have used the fact that, due to the translational symmetry of the lattice, the energy density in cell $C_i$ resulting from a single fluxtube at $\mathbf{x} = \mathbf{0}$ is equivalent to the energy density in cell $C_0$ resulting from a single fluxtube at $\mathbf{x} = \mathbf{x}_{-i}$.

We now apply assumptions 1 and 2 stated above. Within cell $C_0$, we thus approximate $\Psi^{(\text{all})} \simeq \Psi^{(0)} + \delta\Psi^{(i \neq 0)}$, and $\Psi^{(i)} \simeq \Psi^{(\text{none})} + \delta\Psi^{(i)}$ for each $i \neq 0$, where $\delta\Psi^{(i \neq 0)}$ represents the perturbation produced by all fluxtubes external to $C_0$ and $\Psi^{(\text{none})}$ is the Meissner solution. Furthermore, we approximate $\delta\Psi^{(i \neq 0)} \simeq \sum_{i \neq 0} \delta\Psi^{(i)}$ *on the boundary* of $C_0$. We emphasize that $\delta\Psi^{(i)}$ is the linear perturbation to the Meissner solution in the presence of a single fluxtube. In practice, this can be calculated from the Jacobi equations, as we demonstrated in Section 4.2. Under these approximations, we expand $F^{(\text{all})}$ in Equation (A10) and cancel the zeroth-order term $F^{(0)}$ with the $i = 0$ term in the sum. Expanding the remaining $i \neq 0$ contributions and keeping in mind that the Meissner solution has $F = 0$, we are left with an expression for the interaction energy that only contains second variations of $F$, and can thus be written in terms of the vector field $\mathbf{Q}^{(2)}$:

$$\mathcal{F} - \mathcal{F}_\infty \simeq \iint_{C_0} \left[ \delta^2 F[\delta\Psi^{(i \neq 0)}; \Psi^{(0)}] \right] - \sum_{i \neq 0} \left[ \delta^2 F[\delta\Psi^{(i)}; \Psi^{(\text{none})}] \right] \mathrm{d}x\, \mathrm{d}y$$

$$= \int_{\partial C_0} \left[ \mathbf{Q}^{(2)}[\delta\Psi^{(i \neq 0)}; \Psi^{(0)}] - \sum_{i \neq 0} \mathbf{Q}^{(2)}[\delta\Psi^{(i)}; \Psi^{(\text{none})}] \right] \cdot \mathrm{d}\mathbf{S}$$

$$\simeq \int_{\partial C_0} \left[ \mathbf{Q}^{(2)}[\sum_{i \neq 0} \delta\Psi^{(i)}; \Psi^{(0)}] - \sum_{i \neq 0} \mathbf{Q}^{(2)}[\delta\Psi^{(i)}; \Psi^{(\text{none})}] \right] \cdot \mathrm{d}\mathbf{S} , \tag{A11}$$

where $\partial C_0$ represents the boundary of cell $C_0$, and $\mathrm{d}\mathbf{S}$ is the boundary element on this boundary, with outward normal. The final step is to approximate $\Psi^{(0)} \simeq \Psi^{(\text{none})} + \delta\Psi^{(0)}$ on the boundary of $C_0$, and retain only terms up to second order in $\delta\Psi$; this calculation is straightforward once the functional form of $\mathbf{Q}^{(2)}$ is known. In this way, we can express the

interaction energy as an integral over the boundary of a single Wigner–Seitz cell, and so we do not need to consider the full domain volume to determine the nature of the phase transition at the lower critical field.

Note that our assumptions 1 and 2 generally do not apply to the proton order parameter $\psi_{\mathrm{p}}$, because introducing an additional fluxtube causes a nonlinear change in the phase of the proton condensate throughout the domain. Thus, we cannot apply the above method directly to Equation (12). This is the reason for making the change of variables to the set $\Psi \equiv (f, g, \mathbf{V}, \chi)$, for which assumptions 1 and 2 do hold. Now taking first variations of the free energy $F$ in the form of Equation (A6) and using integration by parts, we find that

$$
\begin{aligned}
\mathbf{Q} = {}& 2\delta f \boldsymbol{\nabla} f + \frac{2}{\epsilon}\delta g \boldsymbol{\nabla} g + \frac{2}{\epsilon}g^2 \delta \chi \boldsymbol{\nabla}\chi + 2\kappa^2 \delta\mathbf{V} \times (\boldsymbol{\nabla} \times \mathbf{V}) \\
&+ \frac{2h_1}{\epsilon}\left(g^2 \delta f \boldsymbol{\nabla} f + f^2 \delta g \boldsymbol{\nabla} g + f^2 g^2 \delta\chi(\boldsymbol{\nabla}\chi - \mathbf{V})\right) \\
&+ \frac{2h_2}{\epsilon}fg(\delta f \boldsymbol{\nabla} g + \delta g \boldsymbol{\nabla} f) + 2h_3\left(f^2 \delta f \boldsymbol{\nabla} f + \frac{1}{\epsilon^2}g^2 \delta g \boldsymbol{\nabla} g\right).
\end{aligned}
\tag{A12}
$$

The interaction energy can now be calculated following the steps outlined above. For brevity let us just consider the representative term $\mathbf{Q} = 2fg\,\delta f \boldsymbol{\nabla} g$. For this term, we find that

$$
\mathbf{Q}^{(2)} = \mathbf{Q} + g(\delta f)^2\boldsymbol{\nabla} g + f\,\delta f\,\delta g\boldsymbol{\nabla} g + fg\,\delta f\boldsymbol{\nabla}\delta g.
\tag{A13}
$$

Recalling that the Meissner solution has $f = g = 1$ and $\boldsymbol{\nabla}\chi = \mathbf{V} = \mathbf{0}$, we find that the contribution from this term to the interaction energy (A11) is

$$
\begin{aligned}
&\int_{\partial C_0} \sum_{i \neq 0}\left[2\delta f^{(i)}\boldsymbol{\nabla}\delta g^{(0)} + \sum_{j \neq 0}\delta f^{(i)}\boldsymbol{\nabla}\delta g^{(j)} - \delta f^{(i)}\boldsymbol{\nabla}\delta g^{(i)}\right] \cdot \mathrm{d}\mathbf{S} \\
&= \int_{\partial C_0}\left[\sum_{i,\,j \neq i}\delta f^{(i)}\boldsymbol{\nabla}\delta g^{(j)} + \sum_i\left(\delta f^{(i)}\boldsymbol{\nabla}\delta g^{(0)} - \delta f^{(0)}\boldsymbol{\nabla}\delta g^{(i)}\right)\right] \cdot \mathrm{d}\mathbf{S} \\
&= \sum_i \iint_{C_0}\boldsymbol{\nabla}\cdot\left[\delta f^{(i)}\boldsymbol{\nabla}\delta g^{(0)} - \delta f^{(0)}\boldsymbol{\nabla}\delta g^{(i)}\right]\mathrm{d}x\,\mathrm{d}y.
\end{aligned}
$$

In deriving the last equality, we have used the divergence theorem and the fact that the doubly-summed term vanishes, as can be shown using a similar argument to that leading to Equation (A10):

$$
\begin{aligned}
\sum_i\sum_{j \neq i}\int_{\partial C_0}\left(\delta f^{(i)}\boldsymbol{\nabla}\delta g^{(j)}\right) \cdot \mathrm{d}\mathbf{S} &= \sum_i\sum_{j \neq 0}\int_{\partial C_i}\left(\delta f^{(0)}\boldsymbol{\nabla}\delta g^{(j)}\right) \cdot \mathrm{d}\mathbf{S} \\
&= \sum_{j \neq 0}\int_{\partial\mathbb{R}^2}\left(\delta f^{(0)}\boldsymbol{\nabla}\delta g^{(j)}\right) \cdot \mathrm{d}\mathbf{S} \\
&= 0.
\end{aligned}
$$

Here, $\partial\mathbb{R}^2$ is the boundary of the entire $xy$-plane, where the integrand is exponentially small.

By applying a similar procedure to the remaining terms in Equation (A12), we eventually find that

$$
\mathcal{F} - \mathcal{F}_\infty \simeq \sum_i\iint_{C_0}\left[\delta\Psi^{(i)}\cdot\mathcal{L}[\delta\Psi^{(0)}] - \delta\Psi^{(0)}\cdot\mathcal{L}[\delta\Psi^{(i)}]\right]\mathrm{d}x\,\mathrm{d}y,
\tag{A14}
$$

where $\delta\Psi = (\delta f, \delta g, \delta\mathbf{V}, \delta\chi)$ and

$$
\mathcal{L}[\delta\Psi] = \begin{pmatrix} \left(1 + \dfrac{h_1}{\epsilon} + h_3\right)\nabla^2\delta f + \dfrac{h_2}{\epsilon}\nabla^2\delta g \\[2mm] \dfrac{1}{\epsilon}\left(1 + h_1 + \dfrac{h_3}{\epsilon}\right)\nabla^2\delta g + \dfrac{h_2}{\epsilon}\nabla^2\delta f \\[2mm] -\kappa^2\boldsymbol{\nabla}\times(\boldsymbol{\nabla}\times\delta\mathbf{V}) + \dfrac{h_1}{\epsilon}\boldsymbol{\nabla}\delta\chi \\[2mm] \dfrac{1}{\epsilon}\nabla^2\delta\chi + \dfrac{h_1}{\epsilon}\boldsymbol{\nabla}\cdot(\boldsymbol{\nabla}\delta\chi - \delta\mathbf{V}) \end{pmatrix}. \tag{A15}
$$

We note the similarity between this linear operator $\mathcal{L}[\delta\Psi]$ and the Jacobi Equations (36)–(39). In fact, for an arbitrary free-energy functional $F[\Psi]$ it can be shown that the Formula (A14) for the interaction energy still holds, provided that we define

$$
\mathcal{L}[\delta\Psi] \equiv -\frac{1}{2}\delta^1\left\{\frac{\delta}{\delta\Psi}\langle F\rangle\right\}[\delta\Psi; \Psi^{(\text{none})}], \tag{A16}
$$

i.e., $\mathcal{L}$ is defined by the Jacobi equations, implying that $\mathcal{L}[\delta\Psi^{(i)}] = 0$ at all points except $\mathbf{x} = \mathbf{x}_i$ (the center of the fluxtube), where $\delta\Psi^{(i)}$ is not differentiable. Hence, the integrand in Equation (A14) vanishes at all points inside $C_0$, except at $\mathbf{x} = \mathbf{0}$, where it has the form of a delta function. It is for this reason that the exact location of the fluxtube within the Wigner–Seitz cell is immaterial in Equation (A14).

We can now evaluate the integral in Equation (A14) very much as for the simpler case of a single-component superconductor [54]. In the particular case given by Equation (A15), $\delta\Psi^{(i)}$ is given by Equations (40) and (43), after substituting $r \to |\mathbf{x} - \mathbf{x}_i|$. The terms involving $\delta f$ and $\delta g$ can be evaluated by using the following property of the Bessel function $K_0$:

$$
\nabla^2 K_0(k|\mathbf{x} - \mathbf{x}_{i,0}|) = k^2 K_0(k|\mathbf{x} - \mathbf{x}_{i,0}|) - 2\pi\delta^{(2)}(\mathbf{x} - \mathbf{x}_{i,0}), \tag{A17}
$$

where $\delta^{(2)}$ is the two-dimensional delta function. Using Equations (44) and (45), we then find that all the terms cancel, apart from those involving delta functions, as expected in light of the comments below Equation (A16). The only remaining terms are

$$
(\mathcal{F} - \mathcal{F}_\infty)_{\delta f, \delta g} \simeq
$$

$$
-2\pi \sum_{i\neq 0} K_0\left(\frac{\sqrt{2}|\mathbf{x}_i|}{\xi_j}\right) \sum_{j=1,2}\left[\left(1 + \frac{h_1}{\epsilon} + h_3\right)f_j^2 + 2\frac{h_2}{\epsilon}f_j g_j + \frac{1}{\epsilon}\left(1 + h_1 + \frac{h_3}{\epsilon}\right)g_j^2\right]. \tag{A18}
$$

Again using Equations (44) and (45) to simplify this result, we obtain the second contribution in Equation (46). To evaluate the remaining terms in Equation (A14), it is convenient to use the divergence theorem to rewrite the area integral over $C_0$ as a contour integral over $\partial C_0$ in order to avoid the singular behavior of the Bessel functions at $r = 0$. Remembering that the integrand of Equation (A14) vanishes everywhere but at $\mathbf{x} = \mathbf{0}$, we can shrink the integration contour to a small circle of radius $\varepsilon$ centered around the origin. We then have $d\mathbf{S} = \varepsilon\,d\theta\,\mathbf{e}_r$, and so

$$
(\mathcal{F} - \mathcal{F}_\infty)_{\delta\mathbf{V}} \simeq \kappa^2 \sum_{i\neq 0}\int_{\partial C_0}\left[\delta\mathbf{V}^{(i)}\times(\boldsymbol{\nabla}\times\delta\mathbf{V}^{(0)}) - \delta\mathbf{V}^{(0)}\times(\boldsymbol{\nabla}\times\delta\mathbf{V}^{(i)})\right]\cdot d\mathbf{S}
$$

$$
= -\kappa^2 V_0 \sum_{i\neq 0}\int_0^{2\pi}\varepsilon\left[\delta\mathbf{V}^{(i)}\big|_{\mathbf{x}=0}K_0\left(\frac{\varepsilon}{\lambda_\star}\right)\frac{1}{\lambda_\star} + K_1\left(\frac{\varepsilon}{\lambda_\star}\right)(\boldsymbol{\nabla}\times\delta\mathbf{V}^{(i)})_z\big|_{\mathbf{x}=0}\right]d\theta. \tag{A19}
$$

Using the asymptotic behavior of the Bessel functions, i.e., $K_0(r) \sim -\ln(r)$ and $K_1(r) \sim 1/r$ as $r \to 0$, we observe that in the limit $\varepsilon \to 0$ the first term vanishes, while the second one remains finite. More precisely, we find

$$
\begin{aligned}
(\mathcal{F} - \mathcal{F}_\infty)_{\delta\mathbf{V}} &\simeq -\kappa^2 V_0 \sum_{i\neq 0} \int_0^{2\pi} \lambda_\star (\nabla \times \delta\mathbf{V}^{(i)})_z\big|_{\mathbf{x}=0}\, d\theta \\
&= 2\pi\kappa^2 V_0^2 \sum_{i\neq 0} K_0\left(\frac{|\mathbf{x}_i|}{\lambda_\star}\right),
\end{aligned}
\tag{A20}
$$

the first contribution in Equation (46).

In principle, if we can numerically compute the nonlinear solution for a *single* fluxtube, from this we can determine the values of $V_0$, $f_1$ and $f_2$, and then use Equation (46) to calculate the interaction energy for any lattice of our choosing. In practice, however, it is difficult to obtain both $f_1$ and $f_2$ to sufficiently high accuracy to achieve quantitatively reliable results. Moreover, the assumptions made in obtaining this result are only valid in the asymptotic limit of a widely-spaced lattice, so caution is needed when applying this result to a lattice with finite separation between fluxtubes. For these reasons, we would like to have a more robust method for estimating the interaction energy. Haber and Schmitt [8] have suggested a possible approach: they followed essentially the same steps leading to Equation (A14), but chose to leave the result in the form of an integral over the boundary $\partial C_0$. They then computed this integral numerically, approximating $\delta\Psi^{(i)}$ using the solution obtained numerically for a single fluxtube. However, their approach has a number of shortcomings:

1.  Rather than computing the interaction energy for a lattice, they considered a pair of fluxtubes. However, this is generally not a steady state, i.e., it is not a solution of the Euler–Lagrange Equations (A7). This violates a basic assumption underlying the derivation.
2.  They chose to include some, but not all, of the higher-order terms in their calculation, leading to a result that lacks certain symmetries expected on physical grounds. By contrast, in deriving Equation (A14), we have consistently neglected all terms of higher order than $(\delta\Psi)^2$, and the result is antisymmetric between $\delta\Psi^{(0)}$ and $\delta\Psi^{(i)}$.
3.  Their formula (C8) for the interaction energy depends on the location of the Wigner–Seitz cell boundary, relative to the fluxtube lattice. As we emphasized above, the location of the fluxtube within its cell is immaterial, and therefore should not change the result.

As an alternative approach, we suggest making use of the exact result

$$
\frac{d\mathcal{F}}{d\ln a} = \iint_{C_0} \left[\frac{1}{2}(f^2-1)^2 + \frac{R^2}{2\epsilon}(g^2-1)^2 + \frac{\alpha}{\epsilon}(f^2-1)(g^2-1) - \kappa^2 B_z^2\right] dx\, dy,
\tag{A21}
$$

which we derive in Appendix C. Inside the integral, we can approximate the full solution by superposing the profiles of single fluxtubes with the Meissner solution:

$$
f \simeq 1 + \sum_i (f^{(i)} - 1), \quad g \simeq 1 + \sum_i (g^{(i)} - 1), \quad B_z \simeq \sum_i B_z^{(i)}.
\tag{A22}
$$

This formula is consistent with the rigorous result (46) in the asymptotic limit $a \to \infty$, and because the integrand is spatially periodic by construction, it has none of the shortcomings described above. The formula will be accurate as long as the approximations (A22) hold, which in practice still requires that $a \gg 1$. We have used this formula to independently verify some of the results from our 2D numerical model.

**Appendix C. Weakly Nonlinear Lattice Solution**

We consider a rectangular domain that contains an integer number of fluxtubes, $N$, with area $aN$. We have shown in Section 4.3 that the non-superconducting state is linearly unstable for $a > 2\pi/H_{c2}$, where $H_{c2}$ is given by Equation (49). Our goal is to compute weakly nonlinear solutions for $a = 2\pi/H_{c2} + \delta a$, where $\delta a \ll 1$. To do so we will roughly follow the same procedure as Abrikosov [56], except that by working with a finite domain and quasi-periodic boundary conditions we avoid having to manipulate products of infinite series.

It is convenient at this point to redefine our length scale, so that the area of the domain is normalized to unity, and remains fixed as the parameter $a$ is varied. At the same time, we will also rescale $\mathbf{A}$ so that the boundary conditions have no dependence on $a$. Under the rescaling $\mathbf{x} \to (aN)^{1/2}\mathbf{x}$ and $\mathbf{A} \to (aN)^{-1/2}\mathbf{A}$, the free energy (12) becomes

$$
\begin{aligned}
F[\psi_{\mathrm{p}}, \psi_{\mathrm{n}}, \mathbf{A}] = {} & \frac{1}{2}(1 - |\psi_{\mathrm{p}}|^2)^2 + \frac{R^2}{2\epsilon}(1 - |\psi_{\mathrm{n}}|^2)^2 + \frac{\alpha}{\epsilon}(1 - |\psi_{\mathrm{p}}|^2)(1 - |\psi_{\mathrm{n}}|^2) \\
& + \frac{1}{aN}|(\boldsymbol{\nabla} - \mathrm{i}\mathbf{A})\psi_{\mathrm{p}}|^2 + \frac{1}{\epsilon aN}|\boldsymbol{\nabla}\psi_{\mathrm{n}}|^2 + \frac{\kappa^2}{(aN)^2}|\boldsymbol{\nabla} \times \mathbf{A}|^2 \\
& + \frac{h_1}{\epsilon aN}|(\boldsymbol{\nabla} - \mathrm{i}\mathbf{A})(\psi_{\mathrm{n}}^{\star}\psi_{\mathrm{p}})|^2 + \frac{(h_2 - h_1)}{2\epsilon aN}\boldsymbol{\nabla}(|\psi_{\mathrm{p}}|^2) \cdot \boldsymbol{\nabla}(|\psi_{\mathrm{n}}|^2) \\
& + \frac{h_3}{4aN}\left(\left|\boldsymbol{\nabla}(|\psi_{\mathrm{p}}|^2)\right|^2 + \frac{1}{\epsilon^2}\left|\boldsymbol{\nabla}(|\psi_{\mathrm{n}}|^2)\right|^2\right). \quad\text{(A23)}
\end{aligned}
$$

Our domain now has the dimensions $\Gamma \times (1/\Gamma)$, say, and we have the (quasi)periodic boundary conditions:

$$
\mathbf{A}(\mathbf{x} + \mathbf{L}) = \mathbf{A}(\mathbf{x}) + \pi N \mathbf{e}_z \times \mathbf{L}, \quad\text{(A24)}
$$

$$
\psi_{\mathrm{p}}(\mathbf{x} + \mathbf{L}) = \psi_{\mathrm{p}}(\mathbf{x})\exp(\mathrm{i}\pi N \mathbf{e}_z \times \mathbf{L} \cdot \mathbf{x}), \quad\text{(A25)}
$$

$$
\psi_{\mathrm{n}}(\mathbf{x} + \mathbf{L}) = \psi_{\mathrm{n}}(\mathbf{x}), \quad\text{(A26)}
$$

where $\mathbf{L}$ represents either of the translation symmetries $(\Gamma, 0)$ or $(0, 1/\Gamma)$. The free energy per magnetic flux quantum (in the unscaled units) is

$$
\mathcal{F}(a) = a\overline{F}, \quad\text{(A27)}
$$

where the overbar represents the spatial average, which is equivalent to the area integral over the rescaled rectangular domain. Because the quantity $a$ now appears only as a coefficient in the free energy, we can directly compute the derivative of $\mathcal{F}(a)$, which leads to the formula (A21).

In the rescaled units, the non-superconducting solution has the form $|\psi_{\mathrm{p}}| = 0$, $|\psi_{\mathrm{n}}| = \sqrt{1 + \alpha/R^2}$, $\mathbf{A} = \pi N \mathbf{e}_z \times \mathbf{x}$. At the critical point $a = 2\pi/H_{c2}$, this solution becomes unstable to perturbations $\delta\psi_{\mathrm{p}}$ that lie in the kernel of the linear operator

$$
\mathcal{L} \equiv (\boldsymbol{\nabla} - \mathrm{i}\pi N \mathbf{e}_z \times \mathbf{x})^2 + 2\pi N. \quad\text{(A28)}
$$

These perturbations have the form $\delta\psi_{\mathrm{p}} = \mathrm{e}^{\mathrm{i}\pi N y(x+\mathrm{i}y)}\phi(x + \mathrm{i}y)$, for some function $\phi$, which must be chosen to match the quasi-periodic boundary condition (A25). The general solution can be expressed in terms of Jacobi theta functions:

$$
\phi(z) = \prod_{j=1}^{N}\exp(-2\pi\mathrm{i}y_j z)\,\vartheta_1\left(\frac{\pi}{\Gamma}(z - z_j)\,\middle|\,\frac{\mathrm{i}}{\Gamma^2}\right), \quad\text{(A29)}
$$

where the fluxtube locations, $z_j = x_j + \mathrm{i}y_j$, must satisfy $\sum_j z_j = (m + \frac{1}{2}N)\Gamma + (n + \frac{1}{2}N)\mathrm{i}/\Gamma$, for some $m, n \in \mathbb{Z}$.

Just beyond the critical point, with $a = 2\pi/H_{c2} + \delta a$, we anticipate that the solutions have regular asymptotic expansions of the form

$$\mathbf{A} = \mathbf{A}^{(1)} + (\delta a)\mathbf{A}^{(2)} + \dots, \tag{A30}$$

$$\psi_n = \psi_n^{(1)} + (\delta a)\psi_n^{(2)} + \dots, \tag{A31}$$

$$\psi_p = (\delta a)^{1/2}\psi_p^{(1)} + (\delta a)^{3/2}\psi_p^{(2)} + \dots. \tag{A32}$$

Substituting this ansatz into the rescaled Euler–Lagrange equations, the leading-order contributions recover the non-superconducting state for $\mathbf{A}^{(1)}$ and $\psi_n^{(1)}$, as well as the linear equation $\mathcal{L}\psi_p^{(1)} = 0$. Without loss of generality, we will take $\mathbf{A}^{(1)} = \pi N \mathbf{e}_z \times \mathbf{x}$ and $\psi_n^{(1)} = \sqrt{1 + \alpha/R^2}$. For the moment we do not need to choose the particular form of $\psi_p^{(1)}$. However, in what follows we will make use of its quasi-periodicity, and of the (gauge-invariant) identities

$$\left(\boldsymbol{\nabla} - i\mathbf{A}^{(1)}\right)\psi_p^{(1)} = i\mathbf{e}_z \times \left(\boldsymbol{\nabla} - i\mathbf{A}^{(1)}\right)\psi_p^{(1)}, \tag{A33}$$

$$\frac{1}{2\pi}\nabla^2 \ln|\psi_p^{(1)}| = -N + \sum_j \delta^{(2)}(\mathbf{x} - \mathbf{x}_j), \tag{A34}$$

where $\mathbf{x}_j$ are the fluxtube locations, and $\delta^{(2)}$ is the two-dimensional delta function.

Now proceeding to next order in the Euler–Lagrange equations, we eventually obtain the following:

$$\left(1 - \frac{\alpha^2}{\epsilon R^2}\right)^{-1}\frac{H_{c2}^2\kappa^2}{\pi N}\boldsymbol{\nabla} \times (\boldsymbol{\nabla} \times \mathbf{A}^{(2)}) = \mathbf{e}_z \times \boldsymbol{\nabla}|\psi_p^{(1)}|^2, \tag{A35}$$

$$\left[\left(\frac{1}{|\psi_n^{(1)}|^2} + \frac{h_3}{\epsilon}\right)\nabla^2 - \frac{4\pi N R^2}{H_{c2}}\right]\mathrm{Re}\left\{\psi_n^{(1)\star}\psi_n^{(2)}\right\}$$
$$= -\left[\frac{h_2 - h_1}{2}\nabla^2 - 2\pi N\left(h_1 + \frac{\alpha}{H_{c2}}\right)\right]|\psi_p^{(1)}|^2, \tag{A36}$$

$$\nabla^2 \mathrm{Im}\left\{\psi_n^{(1)\star}\psi_n^{(2)}\right\} = 0, \tag{A37}$$

as well as a lengthy equation for $\psi_p^{(2)}$:

$$\left(1 - \frac{\alpha^2}{\epsilon R^2}\right)\frac{1}{H_{c2}}\mathcal{L}\psi_p^{(2)} = N\left(\frac{2\pi}{H_{c2}}|\psi_p^{(1)}|^2 - 1 + \frac{\alpha^2}{\epsilon R^2}\right)\psi_p^{(1)} - \frac{h_3}{2}\psi_p^{(1)}\nabla^2(|\psi_p^{(1)}|^2)$$
$$-\psi_p^{(1)}\left(\frac{h_2 - h_1}{\epsilon}\nabla^2 - \frac{\alpha}{\epsilon}\frac{4\pi N}{H_{c2}}\right)\mathrm{Re}\left\{\psi_n^{(1)\star}\psi_n^{(2)}\right\}$$
$$+\left(1 - \frac{\alpha^2}{\epsilon R^2}\right)\frac{1}{H_{c2}}\left[(\boldsymbol{\nabla} - i\mathbf{A}^{(1)}) \cdot (i\mathbf{A}^{(2)}\psi_p^{(1)}) + i\mathbf{A}^{(2)} \cdot (\boldsymbol{\nabla} - i\mathbf{A}^{(1)})\psi_p^{(1)}\right]$$
$$-\frac{h_1}{\epsilon}\left[\psi_n^{(1)\star}\psi_n^{(2)}(\boldsymbol{\nabla} - i\mathbf{A}^{(1)})^2\psi_p^{(1)} + (\boldsymbol{\nabla} - i\mathbf{A}^{(1)})^2(\psi_p^{(1)}\psi_n^{(1)}\psi_n^{(2)\star})\right]. \tag{A38}$$

Equation (A35) can be integrated once to obtain $B_z^{(2)}$. We note that the boundary condition (A24) implies that $\mathbf{A}^{(n)}$ is spatially periodic for all $n > 1$, and so $\overline{B_z^{(n)}} = 0$. Hence we find that

$$B_z^{(2)} = \left(1 - \frac{\alpha^2}{\epsilon R^2}\right)\frac{\pi N}{H_{c2}^2\kappa^2}\left(\overline{|\psi_p^{(1)}|^2} - |\psi_p^{(1)}|^2\right). \tag{A39}$$

Now, in order for Equation (A38) to have regular solutions, its right-hand side must be orthogonal to $\psi_{\mathrm{P}}^{(1)}$. To prove this, we note that $\mathcal{L}$ is self-adjoint with respect to the inner product

$$\langle \psi, \phi \rangle \equiv \int_{x=0}^{\Gamma} \int_{y=0}^{1/\Gamma} \psi^\star \phi \, \mathrm{d}y \, \mathrm{d}x \,, \tag{A40}$$

provided that *both* arguments satisfy the quasi-periodic boundary condition (A25). All of the terms $\psi_{\mathrm{P}}^{(n)}$ satisfy these boundary conditions, and so

$$\langle \psi_{\mathrm{P}}^{(1)}, \mathcal{L}\psi_{\mathrm{P}}^{(2)} \rangle = \langle \mathcal{L}\psi_{\mathrm{P}}^{(1)}, \psi_{\mathrm{P}}^{(2)} \rangle = 0 \,.$$

Hence, taking the inner product of $\psi_{\mathrm{P}}^{(1)}$ with Equation (A38), we obtain the compatibility condition

$$0 = N\left( \frac{2\pi}{H_{\mathrm{c2}}} \overline{|\psi_{\mathrm{P}}^{(1)}|^4} - \left(1 - \frac{\alpha^2}{\epsilon R^2}\right) \overline{|\psi_{\mathrm{P}}^{(1)}|^2} \right) + \frac{h_3}{2} \overline{|\boldsymbol{\nabla}(|\psi_{\mathrm{P}}^{(1)}|^2)|^2}$$
$$+ \frac{h_2 - h_1}{\epsilon} \overline{\boldsymbol{\nabla}(|\psi_{\mathrm{P}}^{(1)}|^2) \cdot \boldsymbol{\nabla}\mathrm{Re}\{\psi_{\mathrm{n}}^{(1)\star}\psi_{\mathrm{n}}^{(2)}\}} + \frac{\alpha}{\epsilon} \frac{4\pi N}{H_{\mathrm{c2}}} \overline{|\psi_{\mathrm{P}}^{(1)}|^2 \mathrm{Re}\{\psi_{\mathrm{n}}^{(1)\star}\psi_{\mathrm{n}}^{(2)}\}}$$
$$+ \left(1 - \frac{\alpha^2}{\epsilon R^2}\right) \frac{1}{H_{\mathrm{c2}}} \overline{|\psi_{\mathrm{P}}^{(1)}|^2 B_z^{(2)}} + 4\pi N \frac{h_1}{\epsilon} \overline{|\psi_{\mathrm{P}}^{(1)}|^2 \mathrm{Re}\{\psi_{\mathrm{n}}^{(1)\star}\psi_{\mathrm{n}}^{(2)}\}} \,. \tag{A41}$$

Using Equation (A34), it can be shown that

$$\overline{|\boldsymbol{\nabla}(|\psi_{\mathrm{P}}^{(1)}|^2)|^2} = 2\pi N \overline{|\psi_{\mathrm{P}}^{(1)}|^4} \,, \tag{A42}$$

and combining this with Equations (A36) and (A39), we can write the compatibility condition (A41) in a more symmetric form:

$$\left(1 - \frac{\alpha^2}{\epsilon R^2}\right) \frac{1}{\overline{|\psi_{\mathrm{P}}^{(1)}|^2}} = \left(\frac{2\pi}{H_{\mathrm{c2}}} + \pi h_3\right)\beta + \left(1 - \frac{\alpha^2}{\epsilon R^2}\right)^2 \frac{\pi}{H_{\mathrm{c2}}^3 \kappa^2}(1 - \beta)$$
$$- \frac{R^2}{\epsilon}\left[\frac{1}{2N}\left(\frac{1}{R^2 + \alpha} + \frac{h_3}{\epsilon R^2}\right)\overline{|\boldsymbol{\nabla}\gamma|^2} + \frac{2\pi}{H_{\mathrm{c2}}}\overline{\gamma^2}\right] \,, \tag{A43}$$

where we have defined

$$\beta \equiv \frac{\overline{|\psi_{\mathrm{P}}^{(1)}|^4}}{\left(\overline{|\psi_{\mathrm{P}}^{(1)}|^2}\right)^2} \quad \text{and} \quad \gamma(\mathbf{x}) \equiv \frac{2\mathrm{Re}\{\psi_{\mathrm{n}}^{(1)\star}\psi_{\mathrm{n}}^{(2)}\}}{\overline{|\psi_{\mathrm{P}}^{(1)}|^2}} \,. \tag{A44}$$

If we now expand the free-energy density (A23) up to $O(\delta a^2)$, and use the identities derived above, we eventually find

$$\mathcal{F} = a\overline{F} = \frac{a}{2}\left(1 - \frac{\alpha^2}{\epsilon R^2}\right)\left[1 - \frac{H_{\mathrm{c2}}(\delta a)^2}{2\pi}\overline{|\psi_{\mathrm{P}}^{(1)}|^2}\right] + \frac{(2\pi\kappa)^2}{a} + O(\delta a^3) \,. \tag{A45}$$

The transition to the non-superconducting state is second order if (and only if) $\mathcal{F}(a)$ is convex in a neighborhood of the critical point $a = 2\pi/H_{\mathrm{c2}}$, which requires that

$\mathcal{F}' < 0$ and $\mathcal{F}'' > 0$. Using Equation (A45), and the compatibility condition (A43), these criteria become

$$(H_{c2}\kappa)^2 > \frac{1}{2}\left(1 - \frac{\alpha^2}{\epsilon R^2}\right), \quad (A46)$$

$$\left[\frac{2}{H_{c2}} + h_3 - \left(1 - \frac{\alpha^2}{\epsilon R^2}\right)^2 \frac{1}{H_{c2}^3\kappa^2}\right]\frac{\pi\beta\epsilon}{R^2} > \frac{1}{2N}\left(\frac{1}{R^2 + \alpha} + \frac{h_3}{\epsilon R^2}\right)\overline{|\boldsymbol{\nabla}\gamma|^2} + \frac{2\pi}{H_{c2}}\overline{\gamma^2}. \quad (A47)$$

In a single-component Ginzburg–Landau superconductor these criteria both reduce to $\kappa > 1/\sqrt{2}$, so the upper transition in a type-II superconductor is always second-order [56]. Moreover, minimizing the free energy is equivalent to minimizing $\beta$, and hence the hexagonal lattice is energetically preferred [57]. In our more complicated two-component system, the second criterion (A47) cannot be evaluated analytically. However, for any particular fluxtube arrangement, we can in principle compute $\psi_{\mathrm{P}}^{(1)}$, then solve Equation (A36) to obtain $\gamma$, and thus test this criterion numerically. In particular, for the case of a square or hexagonal lattice, Equation (A36) can be solved in Fourier space, eventually leading to the result (50). We note that the perturbation to the neutron condensate produced by the fluxtubes, which is represented by $\gamma$ in Equation (A47), always acts to reduce the overall free energy, making a first-order transition more likely. The magnitude of $\gamma$ depends, via Equation (A36), on the coupling parameters $h_1$ and $h_2$, and so a first-order transition is guaranteed if these parameters are sufficiently large.

Interestingly, there are certain combinations of the parameters for which Equation (A36) can be solved analytically. In particular, if $\alpha = 0$ and

$$\frac{h_2}{h_1} - 1 = \frac{1}{R^2}\frac{1 + \frac{h_3}{\epsilon}}{1 + \frac{h_1}{\epsilon}}, \quad (A48)$$

then we find that $\gamma \propto |\psi_{\mathrm{P}}^{(1)}|^2$. In that case, the $O(\delta a^2)$ term in the free energy (A45) has the form

$$\left[2\pi\left(1 + \frac{h_1}{\epsilon}\right) + \pi h_3 - \frac{\pi}{\kappa^2}\left(1 + \frac{h_1}{\epsilon}\right)^3 - \frac{1}{\epsilon R^2}\frac{h_1^2}{1 + \frac{h_1}{\epsilon}}\left(1 + \frac{1}{2R^2}\frac{1 + \frac{h_3}{\epsilon}}{1 + \frac{h_1}{\epsilon}}\right)\right]\beta. \quad (A49)$$

If one of the parameters $\epsilon$, $R$ or $\kappa$ is sufficiently small, then the quantity in square brackets will be negative. Taken at face value, this result seems to suggest that the free energy can be made arbitrarily small, because $\beta$ can be made arbitrarily large in an unbounded domain. However, this singularity actually just reflects the breakdown of our weakly nonlinear analysis in the limit of an infinite domain. To illustrate how this breakdown occurs, we have calculated the free energy of various multiply-charged hexagonal and square lattice states for one particular set of parameters, chosen such that the quantity in Equation (A49) is slightly negative. In Figure A1, we plot the resulting free energy as a function of $a$, for several different lattice types. As the value of $a$ is reduced from 14 towards the critical value $2\pi/H_{c2} \simeq 10.3$, we find that the singly-charged hexagonal lattice is replaced by the doubly-charged hexagonal lattice, and then by the triply-charged square lattice, as the energetically preferred lattice state. As $a$ is further reduced, we expect that even more highly-charged lattice states (with higher values of $\beta$) will become energetically preferred. However, throughout this whole range of $a$, the true ground state for an unbounded domain is not a lattice at all, and is instead a mixture of the non-superconducting state and a singly-charged hexagonal lattice. Our weakly-nonlinear analysis does not apply to the true ground state, which develops as a nonlinear instability for $a > 2\pi/H_{c2'} \simeq 9.9$. In fact, for all parameter ranges we have studied, the fluxtubes always adopt a singly-charged, hexagonal pattern whenever the domain is sufficiently large, as seen in Figure 4.

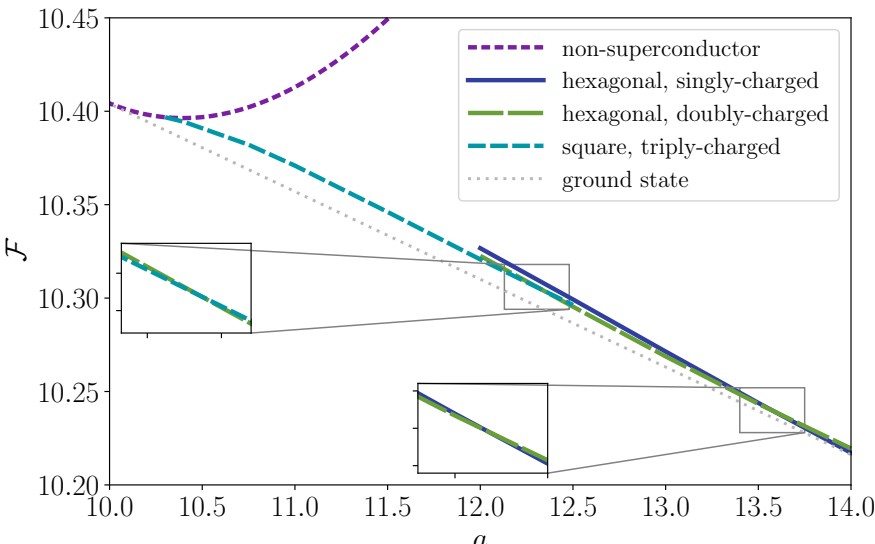

**Figure A1.** The Helmholtz free energy per flux quantum per unit length, $\mathcal{F}(a)$, for various lattice states, with (dimensionless) parameters $h_1 = 0.061$, $h_2 = 0.422$, $h_3 = 0.062$, $R = 0.412$, $\epsilon = 0.096$, and $\kappa = 1.17$. The true ground state (dotted, gray line) arises as a subcritical bifurcation from the non-superconducting state (short-dashed, purple line). The two insets show more detail of the cross-over regions between two lattice configurations of different charge.

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
