# Peer review of "Superconducting Phases in Neutron Star Cores"

_universe, doi:10.3390/universe8040228_

Round 1

Reviewer 1 Report

Referee report on the manuscript "Superconducting Phases in Neutron Star Cores" by Toby S. Wood and  Vanessa Graber.

This work studies the properties of the mixture of condensates of
neutrons and protons in the outer cores of neutron stars, where they coexist using the Ginzburg-Landau (hereafter GL) phenomenological theory. My comments and suggestions are listed below.

1. Introduction, line 12: "...neutrons and protons are both expected to form Cooper pairs, which then condense... ". This statement is confusing, because the formation is Cooper pairs is by itself already the phenomenon of superfluidity or superconductivity in fermionic systems. One can say that the superfluid (or superconductor) is a condensate of Cooper pairs, but one cannot separate the formation of Cooper pairs from the formation of a condensate - it is all the same process. 

2. Introduction, line 15. "...the macroscopic dynamics of the star’s
magnetic field is ultimately determined by the microscopic
interactions between these fluxtubes.." This statement is confusing. First, the crustal magnetic field and its decay via Ohmic dissipation under the additional influence of Hall drift and ambipolar diffusion can control the time-evolution of the magnetic field of a neutron stars. This has little to do with the core superconductor. Secondly, the interaction between flux-tubes is not essential: the key issue is their spatial distribution and their dynamics (for example expulsion) which controls the changes of the total flux of magnetic field passing
through the core. And this is does not depend on the interaction between the proton flux tubes in an important way. 

3. Introduction, line 17.  Ref. [1] claims that the core is a type-II
superconductor - period. Type-I superconductivity was first considered by D. M. Sedrakian et al in Mon. Not. Roy. Astron. Soc. 290 (1997) 203-207 who showed, among other things, that the GL parameter at high densities corresponds to a type-I superconductor and not to type-II as claimed in Ref. [1] and the following literature up to that point.

4. Introduction line 21: The parameter \kappa on line 22 is known as Ginzburg-Landau parameter and it should be said so already here (this is done much later). It would be useful to say here that \kappa > 1/\sqrt{2} corresponds to type-II behavior and kappa <
1/\sqrt{2} to type-I (at least in a single component superconductor). The authors mention only the transition value. 

5. Introduction lines 25-33. The idea of flux bundles was proposed much earlier than [5] or [6], see Astrophys. J. v.447, p.305 by Sedrakian, A. D. and Sedrakian, D. M.  Their treatment uses, in fact, the minimization of Gibbs free-energy as in the second thought experiment of the present authors. Note also that Buckley et al impose isospin symmetry on nuclear interactions which are known to be badly broken (S-wave neutron-proton interaction contains a strong 3S1-3D1 component which is absent for neutron-neutron and proton-proton
interactions). Therefore, their analysis does not apply in the context it was intended to.

6. Introduction, line 39. The notion of the "type-1.5" superconductivity is confusing in the following sense. At least for
one-component superconductors, type-I case has positive surface energy between superconducting and normal regions. In the type-II case, the surface energy is negative. This is the microscopic underpinning of type-I and type-II distinction in the phenomenological GL theory. What is the sign of surface energy in the case of "type-1.5" superconductivity?

7. Introduction, line 43. The impossibility of precession under very
general conditions (strongly coupling between superfluid and normal matter) has been shown in much detail by Sedrakian et al,
Astrophys. J. vol. 524, 341, 1998, before Link's work.  His
contribution is to point out that type-I superconductivity can resolve the tension between the precession (which is not unequivocally observed) and interior coupling between the superfluid and the normal component. The discussion of precession needs to accurately reflect the literature. 

8. I do not have the time to check the computational details of the main body of the paper, but expect that the corrections that are found in this work due to the so-called "consistent" treatment of the GL functional at the scales of the order of the vortex core (around which the GL theory breaks down anyway) are numerically small corrections to those found in the
previous work. For example, how different are the values of H_c2 or H_c1 in the present work compared to the earlier studies?

Author Response

We thank the referee for their careful and considered comments on our work.
Detailed responses to their specific comments and questions are provided below.  Changes to the text have been highlighted in purple in the revised manuscript.

--------------------

Comment 1: Introduction, line 12: "...neutrons and protons are both expected to form Cooper pairs, which then condense... ". This statement is confusing, because the formation is Cooper pairs is by itself already the phenomenon of superfluidity or superconductivity in fermionic systems. One can say that the superfluid (or superconductor) is a condensate of Cooper pairs, but one cannot separate the formation of Cooper pairs from the formation of a condensate - it is all the same process.

Response 1: The referee is right to say that what we had written here was a little imprecise, because pairing and condensation are in reality aspects of the same process.  Moreover, the Cooper pairs are constantly separating and reforming, and cannot therefore be regarded as particles in the usual sense.  We have rephrased the relevant sentence (on line 13) to hopefully avoid misleading the reader.

--------------------

Comment 2: Introduction, line 15. "...the macroscopic dynamics of the star’s magnetic field is ultimately determined by the microscopic interactions between these fluxtubes.." This statement is confusing. First, the crustal magnetic field and its decay via Ohmic dissipation under the additional influence of Hall drift and ambipolar diffusion can control the time-evolution of the magnetic field of a neutron stars. This has little to do with the core superconductor. Secondly, the interaction between flux-tubes is not essential: the key issue is their spatial distribution and their dynamics (for example expulsion) which controls the changes of the total flux of magnetic field passing through the core. And this is does not depend on the interaction between the proton flux tubes in an important way.

Response 2: It is true that the field in the star's crust (where fluxtubes are not present) is governed by other processes.  We have therefore added the word 'core' on line 15 to emphasize that here we are only concerned with the field in the star's core.  However, we would disagree slightly with the referee's second point: it is the interaction between fluxtubes that is responsible for their spatial distribution and dynamics, at least on the scale of the fluxtubes themselves.  For example, it is the mutual attraction/repulsion between fluxtubes that determines the type of superconductivity.  We think that the referee may have simply misinterpreted what we meant by 'interaction' here, so we have rephrased this sentence (on line 16) to hopefully remove any ambiguity.

--------------------

Comment 3:  Introduction, line 17. Ref. [1] claims that the core is a type-II superconductor - period. Type-I superconductivity was first considered by D. M. Sedrakian et al in Mon. Not. Roy. Astron. Soc. 290 (1997) 203-207 who showed, among other things, that the GL parameter at high densities corresponds to a type-I superconductor and not to type-II as claimed in Ref. [1] and the following literature up to that point.

Response 3: The referee is correct that, at the time Ref. [1] was written, estimates for the Ginzburg-Landau parameter in neutron stars suggested type-II superconductivity throughout the core.  Moreover, uncertainties in the equation of state at very high densities meant that most studies focussed on the outer core.  To our knowledge, the first paper to suggest that there might be a type-I inner core was that of Mendell (1991, ApJ 380, 515).  We have therefore added a citation to this paper and the one mentioned by the referee on line 16.

--------------------

Comment 4: 4. Introduction line 21: The parameter \kappa on line 22 is known as Ginzburg-Landau parameter and it should be said so already here (this is done much later). It would be useful to say here that \kappa > 1/\sqrt{2} corresponds to type-II behavior and kappa < 1/\sqrt{2} to type-I (at least in a single component superconductor). The authors mention only the transition value.

Response 4: We have added the definition of the Ginzburg-Landau parameter and the respective regimes on lines 22-23.

--------------------

Comment 5: Introduction lines 25-33. The idea of flux bundles was proposed much earlier than [5] or [6], see Astrophys. J. v.447, p.305 by Sedrakian, A. D. and Sedrakian, D. M. Their treatment uses, in fact, the minimization of Gibbs free-energy as in the second thought experiment of the present authors. Note also that Buckley et al impose isospin symmetry on nuclear interactions which are known to be badly broken (S-wave neutron-proton interaction contains a strong 3S1-3D1 component which is absent for neutron-neutron and proton-proton interactions). Therefore, their analysis does not apply in the context it was intended to.

Response 5: On lines 40-44 we have added a citation to the suggested paper and to another that also considered mechanisms that might produce bundles of fluxtubes.  We agree with the referee's assessment of the work of Buckley et al., but because similar comments were already made in the paper we cite by Alford et al. we prefer not to go into more detail here.

--------------------

Comment 6:  Introduction, line 39. The notion of the "type-1.5" superconductivity is confusing in the following sense. At least for one-component superconductors, type-I case has positive surface energy between superconducting and normal regions. In the type-II case, the surface energy is negative. This is the microscopic underpinning of type-I and type-II distinction in the phenomenological GL theory. What is the sign of surface energy in the case of "type-1.5" superconductivity?

Response 6: We agree that the nomenclature of 'type-1.5' superconductivity has shortcomings, but this has become standard terminology in the literature (e.g. see the review chapter by Babaev et al. in the book Superconductors at the Nanoscale).  In a type-1.5 superconductor, the surface energy (i.e. the energy of the interface between superconducting and normal regions) is generally negative.  However, there is no reason in general why the transition between different types of superconductivity must occur at the point where this surface energy is zero.  We have added some comments about this to the discussion on lines 398-408.  (In fact, we have some preliminary calculations showing that the transition between the type-I and type-1.5 regimes for the case shown in Figure 5 does *not* occur where the surface energy is zero.  These results are not yet ready for publication, however.)

--------------------

Comment 7: Introduction, line 43. The impossibility of precession under very general conditions (strongly coupling between superfluid and normal matter) has been shown in much detail by Sedrakian et al, Astrophys. J. vol. 524, 341, 1998, before Link's work. His contribution is to point out that type-I superconductivity can resolve the tension between the precession (which is not unequivocally observed) and interior coupling between the superfluid and the normal component. The discussion of precession needs to accurately reflect the literature.

Response 7: We have rephrased this paragraph and added a reference to Sedrakian et al. (1999) to address the referee's comment (see lines 47-50).

--------------------

Comment 8: I do not have the time to check the computational details of the main body of the paper, but expect that the corrections that are found in this work due to the so-called "consistent" treatment of the GL functional at the scales of the order of the vortex core (around which the GL theory breaks down anyway) are numerically small corrections to those found in the previous work. For example, how different are the values of H_c2 or H_c1 in the present work compared to the earlier studies?

Response 8: It would be interesting to directly compare the results of our model (which satisfies Galilean invariance) with the model of Haber & Schmitt (which does not).  Unfortunately, in nearly all of their results the entrainment parameter (called g in their notation) is set to zero.

In our discussion section, we compare the critical value of the GL parameter we obtain in our model with the formula used in several earlier works (see equations 52 and 53).  As we now explain on lines 396-398, for realistic choices of the coupling parameters the critical value is significantly larger than predicted by equation (52).

Reviewer 2 Report

The authors present an important revision of the standard picture of the superconductivity of neutron stars.

Previously it was always assumed that for a neutron star superconductor, there is a single critical \kappa parameter \kappa=1/sqrt{2} and single coherence lengths.

The authors base the theory on the observation that inter-component coupling hybridizes protonic and neutronic condensates, and hence protonic condensate must acquire two correlation lengths.

This observation is fully correct.

Then the authors show that for reasonable parameters, the neutrons stars to fall into type-1.5 regime where one coherence length is larger, and another is smaller than magnetic field penetration length.

This is a very valuable and insightful contribution.

However, before recommending the paper the authors need to improve the following:

  1. Introduction

In the introduction, the authors refer to some old articles on multiband superconductors as a "debate", namely: "although the origin of this behavior in the laboratory context remains under debate [12 –15 ]." That is very misleading. The more-than-a-decade-old Refs 12 and 13 were published before the microscopic theory of this state was developed and since that time, all these questions were resolved.

The Ref 12, even at the GL level, is based on a simple mathematical error: explained in section 4.10 in https://library.oapen.org/handle/20.500.12657/27411 .

. The reference 15 has a different kind of a serious issue: it presents fictitious claims of "solutions" of equations that were proven to have no solutions https://aip.scitation.org/doi/full/10.1063/5.0063874 .

It is not correct to claim a debate based on these references, and also, as far as I know, there were no such debates in recent years.

Body of the paper:

The authors write the effective model phenomenologically. Nonetheless could the authors provide a comment on possible microscopic origin of the last term on fourth line of Eq 9 ?

In my opinion, the authors make way too strong reservations regarding the inapplicability of the GL model at low temperatures. That would be correct to make for microscopic models but phenomenological GL models work relatively well even far from Tc even for multicomponent case Physical Review B 85 (13), 134514. Of course, the phenomenological model misses effects such as Kramer-Pesch effect

but I do not see why that would matter too much in this case of a neutron star.

Author Response

We thank the referee for their careful and considered comments on our work.
Detailed responses to their specific comments and questions are provided below.  Changes to the text have been highlighted in purple in the revised manuscript.

--------------------

Comment 1: In the introduction, the authors refer to some old articles on multiband superconductors as a "debate", namely: "although the origin of this behavior in the laboratory context remains under debate [12 –15 ]." That is very misleading. The more-than-a-decade-old Refs 12 and 13 were published before the microscopic theory of this state was developed and since that time, all these questions were resolved. The Ref 12, even at the GL level, is based on a simple mathematical error: explained in section 4.10 in https://library.oapen.org/handle/20.500.12657/27411. The reference 15 has a different kind of a serious issue: it presents fictitious claims of "solutions" of equations that were proven to have no solutions https://aip.scitation.org/doi/full/10.1063/5.0063874. It is not correct to claim a debate based on these references, and also, as far as I know, there were no such debates in recent years

Response 1: We thank the referee for updating us on the subject of multiband superconductors.  We have removed reference to a 'debate' on the issue of type-1.5 superconductivity in laboratory systems and citation of the erroneous papers (see line 37).

--------------------

Comment 2: The authors write the effective model phenomenologically. Nonetheless could the authors provide a comment on possible microscopic origin of the last term on fourth line of Eq 9?

Response 2: As with earlier works on this problem, our model is only intended to be a phenomenological description of the condensates, so we have not attempted to relate our model parameters to microscopic interactions between protons and neutrons.  But our interaction term involving the density gradients can be regarded as a finite-range correction to the effective attraction/repulsion between the condensates (described by the parameter g_pn in equation 9), which itself ultimately arises from the four-fermion neutron-proton interaction.  However, a rigorous microscopic derivation of our model could only be achieved close to the transition temperature, which is not the regime of primary interest here, so we prefer to simply present the model as a phenomenological one.

--------------------

Comment 3: In my opinion, the authors make way too strong reservations regarding the inapplicability of the GL model at low temperatures. That would be correct to make for microscopic models but phenomenological GL models work relatively well even far from Tc even for multicomponent case Physical Review B 85 (13), 134514. Of course, the phenomenological model misses effects such as Kramer-Pesch effect but I do not see why that would matter too much in this case of a neutron star.

Response 3: We entirely agree with the referee's assessment.  We included this discussion of the limitations of the GL model at low temperatures following earlier criticism of our work.  To avoid underselling the model, we have added a reference to the successes of this approach on lines 89-92.

Round 2

Reviewer 1 Report

Concerning point 3.   Mendell's comment on type-I superconductivity is limited to the following: 

"Otherwise type I superconductivity is favored, and the protons would be in the intermediate state instead of the vortex state. Since (m/m*) ~ 2, the protons could form a type I superconductor, though calculations of the proton gap (e.g., see Chao et al. 1972; Takatsuka 1973; Amundsen & Ostgaard 1985) suggest that T_cp may be larger than 10^9 K by a factor large enough to negate this possibility. Another intriguing possibility is that there is a density at which a transition takes place from type II to type I superconductivity."

This may indeed count as a suggestion of type-I superconductivity in neutron stars without any proof. The accuracy of the coverage of the literature by the authors is impressive.

Concerning point 6.  The authors can keep the notion of 1.5 superconductivity. Nevertheless, personally, I find this notion highly dubious because it does not have microscopic underpinning. Furthermore, it is unclear what the number "1.5" signifies?  Why not type-III superconductivity?

Author Response

We thank the referee again for their comments.  We agree that the terminology used for different types of superconductivity is somewhat arbitrary.  The only meaningful way to identify the type of superconductivity is to describe all of the phases and phase transitions, as in our Figure 5 for example.

Reviewer 2 Report

I think the authors addressed my questions .

Just as a side note: for type-1.5 case the ordinary argument on interface energy obtained from 1d solution is not very informative. This is because interface of vortex cluster is not a 1d problem and the boundary conditions for magnetic field is different. So 1d interface problem does  not contain full information on typology. for example interface energy of vortex cluster boundary is positive, still system has vortices. May be , optionally, the authors can clarify slightly that statement.

Author Response

We thank the referee again for their comments.  We agree that the concept of surface/interface energy becomes more complicated when there are other kinds of interfaces that can arise, such as those illustrated in Figure 4.  The fact that such ground states exist implies that these interfaces must have positive energy, and our measurements of the zero-point energy (i.e. the red dots in Figure 2) confirm this.  But because these states are inherently two-dimensional it is more difficult to give a precise definition of their “surface energy”, so we prefer not to discuss this in the paper.